# Disentangling Consensus and Value-Specific Representations for Controllable Pluralistic Value Alignment in LLMs

Jiankui Zhou [1]   Jing Yao [2]   Xiaoyuan Yi [2]   Peng Zhang [1]   Ning Gu [1]   Zhan Hu [1]   Xing Xie [2]   Tun Lu [1]

## Abstract

With the widespread deployment of large language models (LLMs), aligning model outputs with pluralistic human values has become an important research problem. Recent approaches that train task-specific experts and merge them through parameter aggregation have shown promise for pluralistic alignment. However, these methods often overlook the intrinsic complexity of real-world value data, where multiple correlated value dimensions coexist, resulting in highly similar and entangled expert representations. Consequently, modifying the contribution of one value expert may unintentionally influence other values, limiting fine-grained controllability. To address this issue, we propose DisAlign, a model-merging framework that explicitly decomposes value representations into consensus and value-specific components using an information-geometric perspective. DisAlign first extracts a consensus anchor and subspace to capture shared structure across values, and then applies spectral decomposition to the residual representations to construct disentangled value subspaces. This design enables more precise and independent modulation of multiple values. Experiments on three datasets covering different value frameworks demonstrate that DisAlign consistently improves value disentanglement and achieves more accurate pluralistic value control compared to existing baselines. Our code is available at https://github.com/erzhoujk/DisAlign

[1]Fudan University, Shanghai, China [2]Microsoft Research Asia, Beijing, China. Correspondence to: Jing Yao <jingyao@microsoft.com>, Peng Zhang <zhangpeng_@fudan.edu.cn>, Tun Lu <lutun@fudan.edu.cn>.

*Proceedings of the 43rd International Conference on Machine Learning*, Seoul, South Korea. PMLR 306, 2026. Copyright 2026 by the author(s).

## 1. Introduction

With the widespread deployment of Large Language Models (LLMs) (Dubey et al., 2024; Yang et al., 2025) across real-world scenarios, aligning their outputs with human values has become a critical research direction (Wang et al., 2023; Yao et al., 2023; Wang et al., 2024a). Conventional alignment methods, such as RLHF (Ouyang et al., 2022) and DPO (Rafailov et al., 2023), optimize for averaged and monolithic preferences, overlooking substantial heterogeneity in values across cultures, users and contexts. Consequently, *pluralistic value alignment* has attracted increasing attention (Sorensen et al., 2024b; Zhang et al., 2024), which considers diverse alignment objectives composed of *multiple value dimensions with distinct priority weights*.

Existing approaches to pluralistic alignment can be grouped into three categories. First, *prompting-based methods* guide model outputs using value objective prompts (Guo et al., 2025; Wu et al., 2025). While lightweight and flexible, they rely on strong instruction-following and may degrade on smaller models or complex objectives. To overcome this, the second category fine-tunes *separate LLMs for each alignment objective* (Wang et al., 2024b; Yao et al., 2024; Zhou et al., 2024b), with limited scalability due to the high data and computational costs for combinatorial objectives. In recent years, researchers have sought to achieve both flexibility and scalability in pluralistic alignment. To this end, they have transferred mixture-of-experts (MoE) mechanisms (Liu et al., 2025) or linear parameter merging (Ilharco et al., 2022) to this domain, combining multiple value experts. This approach forms a third category of methods (Liu et al., 2024; Wortsman et al., 2022). However, such a direct transfer does not account for the specific complexities inherent to pluralistic alignment.

The complexity of pluralistic values stems from the coupling among value dimensions. Real-world training data rarely reflect a single, isolated value. Specifically, samples labeled for one value dimension often express multiple related values simultaneously (Wortsman et al., 2022). As illustrated in Figure 1(A), data exhibiting a preference for "Truth" also co-activate related values such as "Care" and "loyalty". Moreover, Figure 1(B) shows that the parameters induced during the training of different value experts

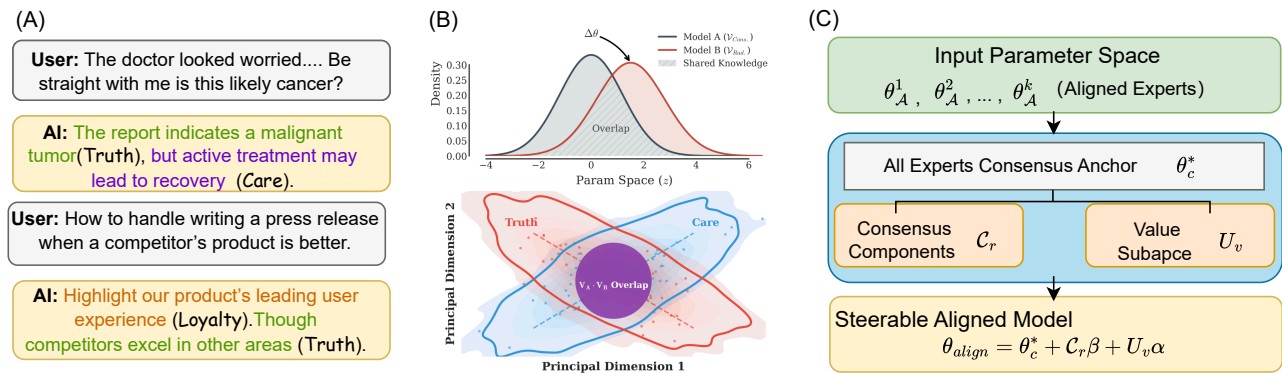

*Figure 1.* (A) Multiple value expressions exist in a single-turn dialogue. (B) Due to the data mixing, parameter coupling occurs. (C) Schematic diagram of our method solving for consensus and unique value subspaces.

exhibit high similarity. The density plot shows that parameter offsets of different value experts heavily overlap in parameter space, indicating substantial shared structure. The PCA plot projects value specific representations into a low-dimensional space, revealing significant overlap between value clusters. Together, they demonstrate the inherent entanglement across value dimensions. Consequently, value experts trained on real-world data fail to isolate individual value dimensions and inevitably exhibit *entangled representations* (Wu et al., 2023). When such entangled experts are merged, adjusting the weight of one expert can unintentionally affect other value dimensions. This interference makes it difficult to control value priorities precisely and independently (Zhu et al., 2024). Therefore, a key challenge is *to separate shared value components from individual value dimensions, enabling precise priority control across multiple values.*

To address this challenge, we propose **DisAlign**, a model-merging framework for controllable pluralistic value alignment, which decomposes entangled experts into two components: i) a consensus component, capturing similarities across experts; ii) a value-specific component, encoding heterogeneous value dimensions. By controlling the weights of the value-specific components, DisAlign enables precise modulation of multiple value dimensions with low cross-value interference. Specifically, DisAlign first extracts value dimension specific information by identifying and removing highly similar consensus components. It then formulates the decomposition in an information-geometric space, which reflects model responses rather than raw parameters.

To compute the consensus component, we construct a consensus anchor by mixing the POE distributions of all experts, capturing global probabilistic consistency. For the consensus subspace, we first compute expert response residuals relative to the consensus anchor. We then perform spectral decomposition on these residuals to extract principal directions with minimal disagreement. These directions

form stable, value dimension specific manifolds that encode general capabilities and shared values. Importantly, these manifolds remain insensitive to other value preferences. Removing the consensus component and applying orthogonalization to each expert yields the encoding of the value-specific component. The final pluralistic model is constructed by linearly combining the consensus anchor, consensus subspace, and user-weighted value vectors. Extensive experiments on three datasets with diverse value frameworks validate the effectiveness of our framework.

In this work, we focus on steerable pluralism (Sorensen et al., 2024b), where the goal is to controllably steer model behavior according to user-specified value combinations. We adopt this setting because it provides explicit and scalable control over value trade-offs while preserving general capabilities. Our framework is particularly applicable in scenarios where predefined value dimensions and corresponding preference data already exist, such as HHH alignment (Askell et al., 2021), constitutional alignment (Vandekerckhove et al., 2017), and recent pluralistic alignment tasks (Ziems et al., 2022; Chiu et al., 2024; Sorensen et al., 2024a). We make the following contributions:

(1) We identify the precise weight control challenge in pluralistic value alignment and propose DisAlign, a novel framework that explicitly decomposes value-shared consensus and value-specific components, enabling disentangled and precise value control.

(2) We formulate this decomposition within an information-geometric framework to capture semantic-level value consensus and ensure low interference value modulation.

(3) Extensive experiments confirm that DisAlign produces more disentangled value representations and achieves significantly more controllable pluralistic value alignment.

## 2. Related Work

### 2.1. Pluralistic Value Alignment

Existing approaches to pluralistic value alignment (Sorensen et al., 2024b; Zhang et al., 2024) can be broadly categorized into three paradigms. The first is *prompting-based methods* (Adams et al., 2025; Guo et al., 2025; Zhong et al., 2025), which steer model outputs through prompting natural language value descriptions (Hu et al., 2024; Sorensen et al., 2024a; Argyle et al., 2023) and illustrative examples (Wu et al., 2025). These methods are lightweight and flexible, but their effectiveness relies on strong instruction-following capabilities and can fail on smaller models and highly complex value objectives. The second category focuses on constructing separate LLMs aligned to individual value objectives via supervised fine-tuning (Shi et al., 2024; Yang et al., 2024; Feng et al., 2024; Rame et al., 2023) or reinforcement learning–based training (Zhou et al., 2024b), including methods such as Value FULCRA (Yao et al., 2024), MODPO (Zhou et al., 2024b), and MAP (Wang et al., 2024b). Pluralistic alignment is typically achieved via joint optimization of multiple reward models, but each new objective combination requires retraining, resulting in high data and computational costs and poor scalability. Balancing flexibility and scalability, an emerging third category first specializes expert models for individual values and combines them through linear parameter merging (Zhu et al., 2024) or automatic mixture-of-experts (MoE) mechanisms, such as PMoL (Liu et al., 2024) and Cultural palette (Yuan et al., 2024) which learn value-specific sub-networks and achieve diversified alignment via gating (Liu et al., 2025). However, these merging mechanisms do not consider the entanglement in value expert models and fail to conduct precise weight control. This paper follows the model-merging paradigm and aims to address this limitation.

### 2.2. Model Merging and Task Vectors

Model merging approaches (Wortsman et al., 2022; Ilharco et al., 2022) modulate LLMs' behaviors by combining parameters linearly. However, simple arithmetic assumes task separability, which falters under conflicting value dimensions. To mitigate this, interference-resolution methods like TIES-Merging (Yadav et al., 2023) and DARE (Yu et al., 2024) employ heuristic sparsification to eliminate conflicting updates. However, these methods treat the gradients of diverse values as interference to be suppressed rather than distinct preferences to be preserved, rendering them unsuitable for pluralistic alignment where maintaining diversity is critical. Recent geometric methods attempt to utilize parameter structure for better aggregation. KnOTS (Stoica et al., 2025) aligns expert subspaces to maximize overlap, while FWA (Matena & Raffel, 2022) enforce orthogonality or use Fisher weighting. However, these methods perform

simplistic decomposition: they enforce orthogonality between tasks without distinguishing value-shared patterns and value-specific characteristics.

## 3. Method

### 3.1. Problem Formulation and Assumptions

Let $p_{\theta_0}(\mathbf{y}|\mathbf{x})$ denote an LLM parameterized by $\theta_0$ that generates a response $\mathbf{y}$ given a prompt $\mathbf{x}$, and $\{v_1, v_2, \ldots, v_k\}$ denote a set of value dimensions that highlight different preferences of people (e.g., care, fairness). Diverse value objectives can be defined as weighted combinations of these value dimensions, $\boldsymbol{v} = \{(v_1, \lambda_1), \ldots, (v_k, \lambda_k)\}$, where $\lambda_i$ specifies the priority weight of value $v_i$. Pluralistic value alignment aims to steer the LLM's outputs to align with any given value objective $\boldsymbol{v}$. For each value dimension $v_i$, we have access to the preference data $\mathcal{D}_i = \{(x, y_1^{(i)} \succ y_2^{(i)})\}$, where $y_1^{(i)}$ is preferred over $y_2^{(i)}$ with respect to value $v_i$. Using all data, we obtain a set of value-aligned expert models $\{p_{\theta_{\mathcal{A}}^{(i)}}(\mathbf{y}|\mathbf{x})\}_{i=1}^k$. Their parameter offsets from the base LLM is computed as $\Delta\theta^{(i)} = \theta_{\mathcal{A}}^{(i)} - \theta_0$ and define $Q = [\Delta\theta^{(1)}, \ldots, \Delta\theta^{(k)}]$. Following the flexible and scalable model-merging paradigm for pluralistic value alignment (see Sec. 2.1), we aim to design an aggregation method over these value experts $\{\theta_{\mathcal{A}}^{(i)}\}_{i=1}^k$ to achieve precise priority control over value dimensions.

Our method relies on two standard assumptions.

**Assumption 3.1** (Local quadratic regime). All individual value experts $\{\theta_{\mathcal{A}}^{(i)}\}_{i=1}^k$ lie in a sufficiently small neighborhood of the base LLM parameters $\theta_0$.

**Assumption 3.2** (Local metric homogeneity). The empirical Fisher information matrices admit a well-conditioned representative metric $\bar{F} = \frac{1}{k}\sum_{i=1}^k \tilde{F}_i \succ 0$, such that the operator norm $|\tilde{F}_i - \bar{F}|_2$ remains small within the region

Assumption 3.1 underlines model merging techniques (Matena & Raffel, 2022; Wortsman et al., 2022) and has been empirically validated in prior work (Zhou et al., 2024a). Assumption 3.2 is supported by (Aghajanyan et al., 2021) and commonly adopted in model merging based alignment methods (Ilharco et al., 2022). Detailed justification and sensitivity analysis of both assumptions are provided in Appendix A.

### 3.2. Overview of DisAlign Framework

As discussed and empirically validated in Sec. 1, the data $\mathcal{D}_i$ for the training of each value expert $\theta_{\mathcal{A}}^{(i)}$ often simultaneously express other correlated values and lead to entangled representation in these experts, limiting precise and independent priority control during merging. To address this challenge, we propose **DisAlign**, a novel framework to

explicitly decompose these entangled experts into *shared consensus components* and *value-specific components* and provide the algorithm 1. Concretely, DisAlign follows a two-stage workflow:

• In the first stage, we characterize distributional consensus in the probability space via a Product-of-Experts (PoE) formulation and obtain a parameter consensus anchor $\theta_c^\star$ through information projection (Section 3.3). We then identify directions of minimal disagreement among experts under the Fisher geometry by solving a generalized eigenvalue problem (GEVP), thereby constructing the consensus subspace $\mathcal{C}_r$ (Section 3.4).

• In the second stage, we remove the projection onto the consensus subspace from each expert and orthogonalize the remaining components to isolate value-specific directions, enabling precise and independent value control without degrading shared capabilities (Section 3.5).

We elaborate on each model in the following subsections.

### 3.3. Consensus Anchor via Information Projection

First, we aim to construct a parameter anchor $\theta_c^\star$ that captures the pattern shared across expert models. Instead of simplistic similarity among these parameters $\{\theta_{\mathcal{A}}^{(i)}\}_{i=1}^k$, we think the shared pattern caused by value alignment should be identified from the similarity among the response generation behaviors of these models. To this end, we introduce the Product-of-Experts (PoE) framework $\tilde{p}_{\text{poe}}(\mathbf{y} \mid \mathbf{x}) \propto \prod_{i=1}^k p_{\theta_{\mathcal{A}}^{(i)}}(\mathbf{y} \mid \mathbf{x})$. (Hinton, 2002; Amari, 2016), (Amari, 1998) and formalize the optimization of the parameter anchor $\theta_c^\star$ as minimizing the KL divergence:

$$\theta_c^\star = \arg\min_\theta D_{\text{KL}}(\tilde{p}_{\text{poe}}\|p_\theta). \qquad (1)$$

As a geometric mean of distributions, it can precisely extract the intrinsic consensus among experts via (I)-projection on the probability manifold. Compared to conventional additive aggregation, PoE provides a geometry anchor that effectively disentangles highly entangled value dimensions, laying a structured foundation for fine-grained control.

While exact optimization of Eq. 1 is intractable, under the local quadratic assumption 3.1, the product of expert distributions corresponds to the sum of their quadratic log-likelihoods, effectively reducing the PoE optimization to a tractable Fisher-weighted least squares problem:

$$\theta_c^\star = \arg\min_\theta \sum_{i=1}^k (\theta - \theta_{\mathcal{A}}^{(i)})^\top F_i(\theta - \theta_{\mathcal{A}}^{(i)}). \qquad (2)$$

This formulation intuitively positions the anchor as the geometric barycenter of the experts, where the contribution of each expert is weighted by its local Fisher information $F_i(\theta - \theta_{\mathcal{A}}^{(i)})$.

Solving Eq. 2 subject to the subspace constraint yields our analytical result:

**Proposition 3.3** (Approximate stationary point solution)**.** *Under Assumption 3.1, the stationary point of the Fisher-weighted quadratic surrogate (Eq. 2) restricted to the basis Q admits the following closed-form approximation:*

$$\theta_{\mathbf{c}}^\star \approx \theta_{\mathbf{0}} + Q \left( \sum_{i=1}^k Q^\top F_i Q \right)^{-1} \sum_{i=1}^k Q^\top F_i(\theta_{\mathcal{A}}^{(i)} - \theta_0). \quad (3)$$

*where*

$$F_i := \mathbb{E}_{\mathbf{x},\mathbf{y}\sim p_{\theta_i}}[\nabla_\theta \log p_{\pi_\theta}(\mathbf{y}|\mathbf{x})\nabla_\theta \log p_{\pi_\theta}(\mathbf{y}|\mathbf{x})^\top]_{\theta=\theta_{\mathbf{i}}}$$

*denotes the expected Fisher information matrix.*

In practice, we replace $F_i$ with its block empirical approximation $\tilde{F}_i$ (Wu et al., 2024). A detailed comparison with other approximation schemes is provided in Appendix E.4. The closed-form solution enables efficient computation without iterative optimization. A detailed proof of Proposition 3.3 is provided in Appendix B

### 3.4. Spectral Decomposition of the PoE-Induced Consensus Geometry

With the parameter anchor $\theta_{\mathbf{c}}^\star$, we characterize alignment directions shared across experts in the PoE-induced tangent space. However, this anchor point only represents global consensus information. We further model important local consensus information by building a consensus subspace.

Let $\mathbf{r}_i = \theta_{\mathcal{A}}^{(\mathbf{i})} - \theta_{\mathbf{c}}^\star \in \mathbb{R}^d$ denote the residual of expert $\theta_{\mathcal{A}}^{(\mathbf{i})}$. The empirical residual covariance is $\Sigma_r = \frac{1}{k}\sum_{i=1}^k \mathbf{r}_i \mathbf{r}_i^\top$. Since $\theta_{\mathbf{c}}^\star$ is obtained from a local quadratic approximation of the KL divergence, residual comparisons must respect the information geometry induced by the PoE objective. Accordingly, the tangent space is equipped with the aggregated block Fisher metric $\sum_{i=1}^k \tilde{F}_i$. Intuitively, we seek directions along which expert disagreement is small relative to the local Fisher curvature, leading to a geometry-aware notion of consensus. For a candidate subspace with basis $B \in \mathbb{R}^{d \times r}$, this is quantified by the Fisher-normalized residual energy

$$\min_B Tr(B^\top \Sigma_r B) \quad \text{s.t.} \quad B^\top \sum_{i=1}^k \tilde{F}_i B = I_r. \qquad (4)$$

To find directions of minimal disagreement, we minimize the Fisher-normalized residual energy, which reduces to the Generalized Eigenvalue Problem (GEVP) in Eq. 5.

$$\Sigma_r \mathbf{b}_j = \lambda_j \left( \sum_{i=1}^k \tilde{F}_i \right) \mathbf{b}_j. \qquad (5)$$

Let $0 \leq \lambda_1 \leq \cdots \leq \lambda_m$. We estimate the effective consensus dimensionality from the GEVP spectrum using an entropy-based criterion. Defining $\sigma_j =$

$1/(\lambda_j + \epsilon)$ and the normalized weights $p_j \propto \sigma_j$, and $r_{\text{eff}} = \exp\left(-\sum_{j=1}^{m} p_j \log p_j\right)$. We choose $r = \lfloor r_{\text{eff}} \rfloor$. Finally, under mild regularity conditions, we build a consensus subspace $\mathcal{C}_r = [\mathbf{b}_1, \ldots, \mathbf{b}_r]$.

**Proposition 3.4** (Stability and Identifiability of the PoE-Induced Consensus Subspace)**.** *Let $\theta_c^\star$ be the PoE anchor and $\mathcal{C}_r$ the subspace spanned by the $r$ smallest generalized eigenvectors of Eq. 5. Under Assumptions 3.1 and 3.2, if a spectral gap $\lambda_r < \lambda_{r+1}$ exists, then: (i) $\mathcal{C}_r$ is uniquely defined (up to orthogonal transformation) and stable to small perturbations of the residual statistics. (ii )Under the local quadratic PoE approximation, $\mathcal{C}_r$ is invariant to global translations of the anchor point, and thus constitutes an identifiable geometric representation of shared alignment across experts.*

Proposition 3.4 shows that the PoE-induced consensus subspace is intrinsically defined and robust, remaining stable under estimation noise and invariant to anchor translations.

### 3.5. Disentangled Value-Specific Subspaces

While consensus subspaces can capture the orientation of alignment parameters shared in large localities, they cannot provide controllable and value-specific orientations. Therefore, we construct a consensus orthogonal residual subspace to achieve adjustable value adjustment.

For each value-aligned expert residual parameter $\mathbf{r}_i = \theta_{\mathcal{A}}^{(\mathbf{i})} - \theta_{\mathbf{c}}^\star \in \mathbb{R}^d$, we first remove its consensus component by projecting it onto the Fisher-orthogonal complement of the consensus subspace:

$$\tilde{\mathbf{v}}^{(i)} := \mathbf{r}_i - \Pi_{\mathcal{C}_r}^F(\mathbf{r}_i), \tag{6}$$

where $\Pi_{\mathcal{C}_r}^F(\cdot)$ denotes the projection operator onto the consensus subspace $\mathcal{C}_r$:

$$\Pi_{\mathcal{C}_r}^F = \mathcal{C}_r\left(\mathcal{C}_r^\top(\sum_{i=1}^{k} \tilde{F}_i)\mathcal{C}_r\right)^{-1}\mathcal{C}_r^\top(\sum_{i=1}^{k} \tilde{F}_i). \tag{7}$$

By construction, each residual $\tilde{\mathbf{v}}^{(i)}$ is Fisher-orthogonal to $\mathcal{C}_r$, ensuring that subsequent value directions correspond to PoE-neutral perturbations.

We stack the consensus-removed residuals into a residual matrix $R := [\tilde{\mathbf{v}}^{(1)}, \ldots, \tilde{\mathbf{v}}^{(k)}] \in \mathbb{R}^{d \times k}$, and characterize their geometry using the Fisher-weighted Gram matrix $G = R^\top\left(\sum_{i=1}^{k} \tilde{F}_i\right)R$. Since the number of values $k$ is typically small compared to the parameter dimension $d$, we solve the associated dual eigenproblem efficiently:

$$G\mathbf{e}_j = \mu_j\mathbf{e}_j, \qquad \|\mathbf{e}_j\|_2 = 1. \tag{8}$$

The corresponding value specific directions are recovered via $\mathbf{u}_j = \frac{1}{\sqrt{\mu_j}}R\mathbf{e}_j$, yielding a Fisher-orthonormal basis

---

**Algorithm 1** The DisAlign Framework

**Require:** Base LLM $\theta_0$, value-aligned experts $\{\theta_{\mathcal{A}}^{(i)}\}_{i=1}^k$, Fisher matrices $\{\tilde{F}_i\}_{i=1}^k$, subspace basis $Q$, and value priority weights $\alpha$.
**Ensure:** Model aligned with the value objective $\theta_{\text{align}}$
1: **Step 1: Consensus Anchor via Information Projection**
2: Compute Fisher Anchor $\theta_{\mathbf{c}}^\star$ via proposition 3.3
3: **Step 2: Spectral Decomposition of the PoE-Induced Consensus Geometry**
4: Residuals coordinates: $\mathbf{r}_i \leftarrow \theta_{\mathcal{A}}^{(i)} - \theta_{\mathbf{c}}^\star$.
5: Solve GEVP for directions $\mathbf{b}_j$ via Eq. 5:
6: Define $\mathcal{C}_r = [\mathbf{b}_1, \ldots, \mathbf{b}_r]$.
7: **Step 3: Disentangled Value-Specific Subspaces**
8: Remove consensus projection:
9:     $\tilde{\mathbf{v}}^{(i)} \leftarrow \mathbf{r}_i - \Pi_{\mathcal{C}_r}^F(\mathbf{r}_i)$
10: Compute value basis $U_v$ via equation 8
11: **Return** Steerable parameters $\theta_{\text{align}} = \theta_{\mathbf{c}}^* + \beta\mathcal{C}_r + \alpha U_v$.

---

$U_v = [\mathbf{u}_1, \ldots, \mathbf{u}_m]$ that spans the adjustable value subspace. Given value control coefficients $\alpha \in \mathbb{R}^m$, the final steered model is expressed as

$$\theta_{\text{align}} = \theta_{\mathbf{c}}^* + \beta\mathcal{C}_r + \alpha U_v. \tag{9}$$

In Eq. 9, the coefficient $\beta$ represents the spectral weights of the consensus principal components, serving as a control vector that can be set to $\mathbf{1}$ to default to the consensus anchor. Let $R = U\Sigma^{1/2}E^\top$ denote the Fisher weighted SVD of the residual matrix. To enable compositional value control, we parameterize value mixtures by $\lambda \in \mathbb{R}^k$ and map them into the Fisher-orthonormal value subspace via $\alpha(\lambda) = \Sigma^{1/2}E^\top\lambda$, such that $U_v\alpha(\lambda) = \sum_{i=1}^{k} \lambda_i\tilde{\mathbf{v}}^{(i)}$, ensuring stable and disentangled value steering.

## 4. Experiments

### 4.1. Experimental Setup

**Datasets** To evaluate our framework across a diverse spectrum of values ranging from fundamental moral intuitions to cognitive styles, we utilize three representative datasets: **MIC** (Ziems et al., 2022), **Daily Dilemmas** (Chiu et al., 2024), and **ValuePrism** (Sorensen et al., 2024a). In the training set, each question is associated with multiple answers driven by different value dimensions, compiled into the SFT training set for value expert models. As for the test set, each sample consists of one question, three candidate answers and three distinct value objectives corresponding to the answers respectively. Detailed data preprocessing is provided in Appendix D.1.

**Evaluation Metrics** Following prior work (Adams et al., 2025), we consider three types of value objectives that re-

*Table 1.* Evaluation of steerable pluralistic alignment on **MIC**, **Daily Dilemmas**, and **ValuePrism**. Metrics follow the Steerable Pluralism framework: **A** – Two-dimensional Accuracy (↑), **B** – Random-dimensional Accuracy Accuracy (↑), **C** – Full-dimensional Accuracy (↑). **Bold** indicates the best per metric. The second best results are indicated with underlining. Results are averaged over three runs with different seeds. ∗ indicates results that exceed the second-best by more than three standard deviations.

| Backbone | Method | MIC (Moral Foundations) | | | Daily Dilemmas (Preference) | | | ValuePrism (Values) | | |
|---|---|---|---|---|---|---|---|---|---|---|
| | | A. Two D. | B. Random D. | C. Full D. | A. Two D. | B. Random D. | C. Full D. | A. Two D. | B. Random D. | C. Full D. |
| Llama-3.2-3B | *Inference-time & Prompting-based Methods* | | | | | | | | | |
| | Steerable Pluralism | $53.2_{\pm0.27}$ | $52.4_{\pm0.25}$ | $53.5_{\pm0.26}$ | $51.5_{\pm0.32}$ | $53.6_{\pm0.31}$ | $51.8_{\pm0.29}$ | $52.4_{\pm0.32}$ | $51.2_{\pm0.28}$ | $52.1_{\pm0.24}$ |
| | Modular Pluralism | $55.8_{\pm0.23}$ | $53.2_{\pm0.34}$ | $52.5_{\pm0.27}$ | $54.3_{\pm0.37}$ | $53.8_{\pm0.34}$ | $52.2_{\pm0.31}$ | $52.8_{\pm0.42}$ | $52.8_{\pm0.39}$ | $53.3_{\pm0.23}$ |
| | Prompt-Align | $\underline{56.1}_{\pm0.29}$ | $\underline{54.3}_{\pm0.34}$ | $\underline{54.2}_{\pm0.31}$ | $\underline{57.9}_{\pm0.29}$ | $\underline{58.7}_{\pm0.33}$ | $\underline{56.1}_{\pm0.35}$ | $\underline{57.4}_{\pm0.20}$ | $52.8_{\pm0.19}$ | $\underline{55.8}_{\pm0.13}$ |
| | *Fine-tuning based Method* | | | | | | | | | |
| | MAP | $54.2_{\pm0.33}$ | $49.8_{\pm0.35}$ | $53.5_{\pm0.38}$ | $53.1_{\pm0.29}$ | $54.2_{\pm0.31}$ | $52.8_{\pm0.32}$ | $52.5_{\pm0.16}$ | $53.1_{\pm0.20}$ | $52.2_{\pm0.19}$ |
| | CPO | $53.7_{\pm0.37}$ | $50.3_{\pm0.33}$ | $52.8_{\pm0.39}$ | $53.8_{\pm0.35}$ | $51.9_{\pm0.36}$ | $52.6_{\pm0.36}$ | $52.9_{\pm0.26}$ | $50.1_{\pm0.31}$ | $49.7_{\pm0.36}$ |
| | *Model-merging based Method* | | | | | | | | | |
| | PAS | $53.5_{\pm0.46}$ | $48.5_{\pm0.42}$ | $53.1_{\pm0.41}$ | $56.2_{\pm0.39}$ | $58.5_{\pm0.36}$ | $54.9_{\pm0.42}$ | $55.1_{\pm0.36}$ | $\underline{56.8}_{\pm0.46}$ | $54.2_{\pm0.38}$ |
| | Pmol | $50.4_{\pm0.45}$ | $50.3_{\pm0.43}$ | $50.5_{\pm0.44}$ | $49.9_{\pm0.34}$ | $53.8_{\pm0.38}$ | $51.5_{\pm0.36}$ | $51.9_{\pm0.39}$ | $52.8_{\pm0.47}$ | $51.1_{\pm0.32}$ |
| | Rewarded Soups | $50.8_{\pm0.51}$ | $42.5_{\pm0.53}$ | $51.2_{\pm0.49}$ | $49.1_{\pm0.42}$ | $41.2_{\pm0.48}$ | $50.5_{\pm0.41}$ | $48.9_{\pm0.39}$ | $40.8_{\pm0.36}$ | $50.1_{\pm0.49}$ |
| | FWA | $51.7_{\pm0.51}$ | $46.3_{\pm0.49}$ | $50.1_{\pm0.43}$ | $48.6_{\pm0.49}$ | $49.1_{\pm0.47}$ | $52.9_{\pm0.43}$ | $50.3_{\pm0.39}$ | $51.9_{\pm0.38}$ | $52.6_{\pm0.41}$ |
| | DARE | $50.5_{\pm0.47}$ | $51.8_{\pm0.41}$ | $51.0_{\pm0.43}$ | $50.5_{\pm0.53}$ | $49.6_{\pm0.51}$ | $50.6_{\pm0.56}$ | $51.2_{\pm0.42}$ | $52.3_{\pm0.42}$ | $50.3_{\pm0.41}$ |
| | AlignMerge | $51.3_{\pm0.46}$ | $51.2_{\pm0.43}$ | $49.3_{\pm0.41}$ | $48.9_{\pm0.51}$ | $47.1_{\pm0.53}$ | $51.6_{\pm0.55}$ | $50.1_{\pm0.45}$ | $48.9_{\pm0.41}$ | $48.7_{\pm0.54}$ |
| | KnOTS | $51.5_{\pm0.45}$ | $49.5_{\pm0.49}$ | $51.8_{\pm0.45}$ | $50.1_{\pm0.49}$ | $49.9_{\pm0.48}$ | $51.2_{\pm0.41}$ | $49.8_{\pm0.43}$ | $43.2_{\pm0.44}$ | $51.9_{\pm0.40}$ |
| | VISPA | $53.2_{\pm0.44}$ | $49.7_{\pm0.47}$ | $53.5_{\pm0.43}$ | $53.4_{\pm0.51}$ | $51.6_{\pm0.56}$ | $51.9_{\pm0.54}$ | $52.6_{\pm0.48}$ | $52.6_{\pm0.45}$ | $51.7_{\pm0.42}$ |
| | **DisAlign(Ours)** | $\mathbf{58.1^*}_{\pm0.53}$ | $\mathbf{59.6^*}_{\pm0.51}$ | $\mathbf{56.6^*}_{\pm0.59}$ | $\mathbf{59.0^*}_{\pm0.49}$ | $\mathbf{60.8^*}_{\pm0.51}$ | $\mathbf{57.1^*}_{\pm0.52}$ | $\mathbf{58.8^*}_{\pm0.46}$ | $\mathbf{58.8^*}_{\pm0.42}$ | $\mathbf{56.2}_{\pm0.43}$ |
| | **p-value** | $< 0.01$ | $< 0.001$ | $< 0.01$ | $< 0.05$ | $< 0.001$ | $< 0.05$ | $< 0.05$ | $< 0.001$ | $< 0.05$ |
| Qwen3.5-4B | *Inference-time & Prompting-based Methods* | | | | | | | | | |
| | Steerable Pluralism | $55.4_{\pm0.26}$ | $53.2_{\pm0.23}$ | $54.9_{\pm0.29}$ | $51.2_{\pm0.34}$ | $54.9_{\pm0.31}$ | $54.2_{\pm0.29}$ | $55.4_{\pm0.38}$ | $54.2_{\pm0.22}$ | $54.8_{\pm0.22}$ |
| | Modular Pluralism | $57.2_{\pm0.22}$ | $56.3_{\pm0.31}$ | $54.2_{\pm0.21}$ | $57.4_{\pm0.35}$ | $55.2_{\pm0.31}$ | $54.3_{\pm0.30}$ | $55.2_{\pm0.38}$ | $54.3_{\pm0.35}$ | $55.8_{\pm0.24}$ |
| | Prompt-Align | $\underline{59.1}_{\pm0.29}$ | $\underline{57.5}_{\pm0.32}$ | $\underline{57.2}_{\pm0.32}$ | $\underline{61.2}_{\pm0.23}$ | $\underline{61.8}_{\pm0.29}$ | $\underline{59.3}_{\pm0.32}$ | $\underline{60.4}_{\pm0.23}$ | $55.3_{\pm0.15}$ | $\underline{58.2}_{\pm0.13}$ |
| | *Fine-tuning based Method* | | | | | | | | | |
| | MAP | $55.5_{\pm0.35}$ | $52.3_{\pm0.32}$ | $55.8_{\pm0.34}$ | $56.7_{\pm0.31}$ | $56.8_{\pm0.36}$ | $55.3_{\pm0.35}$ | $53.9_{\pm0.19}$ | $55.5_{\pm0.23}$ | $54.5_{\pm0.24}$ |
| | CPO | $53.8_{\pm0.33}$ | $51.4_{\pm0.35}$ | $55.2_{\pm0.36}$ | $54.1_{\pm0.34}$ | $56.5_{\pm0.34}$ | $54.7_{\pm0.38}$ | $55.3_{\pm0.23}$ | $53.1_{\pm0.34}$ | $51.8_{\pm0.33}$ |
| | *Model-merging based Method* | | | | | | | | | |
| | PAS | $54.7_{\pm0.41}$ | $49.9_{\pm0.46}$ | $54.9_{\pm0.39}$ | $57.8_{\pm0.36}$ | $60.7_{\pm0.34}$ | $56.7_{\pm0.39}$ | $56.7_{\pm0.38}$ | $\underline{58.3}_{\pm0.38}$ | $57.7_{\pm0.32}$ |
| | Pmol | $52.3_{\pm0.39}$ | $52.6_{\pm0.41}$ | $53.4_{\pm0.48}$ | $51.3_{\pm0.32}$ | $54.3_{\pm0.37}$ | $53.7_{\pm0.41}$ | $54.3_{\pm0.36}$ | $54.3_{\pm0.43}$ | $53.7_{\pm0.35}$ |
| | Rewarded Soups | $51.5_{\pm0.49}$ | $43.2_{\pm0.52}$ | $52.2_{\pm0.47}$ | $50.9_{\pm0.38}$ | $41.9_{\pm0.46}$ | $52.6_{\pm0.43}$ | $51.2_{\pm0.39}$ | $44.3_{\pm0.30}$ | $53.4_{\pm0.42}$ |
| | FWA | $53.4_{\pm0.53}$ | $46.5_{\pm0.44}$ | $53.2_{\pm0.41}$ | $50.6_{\pm0.51}$ | $50.9_{\pm0.39}$ | $54.6_{\pm0.45}$ | $52.8_{\pm0.36}$ | $52.9_{\pm0.32}$ | $55.1_{\pm0.39}$ |
| | DARE | $51.3_{\pm0.42}$ | $50.1_{\pm0.38}$ | $52.9_{\pm0.45}$ | $51.1_{\pm0.45}$ | $44.5_{\pm0.48}$ | $53.1_{\pm0.43}$ | $53.9_{\pm0.39}$ | $47.9_{\pm0.46}$ | $52.8_{\pm0.41}$ |
| | AlignMerge | $53.4_{\pm0.42}$ | $53.8_{\pm0.39}$ | $51.8_{\pm0.43}$ | $50.2_{\pm0.47}$ | $47.4_{\pm0.47}$ | $51.7_{\pm0.42}$ | $52.8_{\pm0.45}$ | $50.1_{\pm0.49}$ | $51.5_{\pm0.52}$ |
| | KnOTS | $54.1_{\pm0.42}$ | $53.8_{\pm0.43}$ | $54.8_{\pm0.42}$ | $52.7_{\pm0.51}$ | $52.8_{\pm0.44}$ | $53.3_{\pm0.47}$ | $52.4_{\pm0.42}$ | $45.2_{\pm0.43}$ | $52.9_{\pm0.32}$ |
| | VISPA | $55.9_{\pm0.42}$ | $52.2_{\pm0.52}$ | $55.8_{\pm0.39}$ | $55.1_{\pm0.48}$ | $54.1_{\pm0.47}$ | $53.4_{\pm0.51}$ | $54.8_{\pm0.40}$ | $55.1_{\pm0.34}$ | $53.5_{\pm0.33}$ |
| | **DisAlign(Ours)** | $\mathbf{60.0}_{\pm0.46}$ | $\mathbf{62.4^*}_{\pm0.46}$ | $\mathbf{58.2}_{\pm0.49}$ | $\mathbf{63.3^*}_{\pm0.47}$ | $\mathbf{62.7}_{\pm0.47}$ | $\mathbf{63.0^*}_{\pm0.41}$ | $\mathbf{61.3^*}_{\pm0.42}$ | $\mathbf{62.4^*}_{\pm0.42}$ | $\mathbf{60.4^*}_{\pm0.33}$ |
| | **p-value** | $< 0.05$ | $< 0.001$ | $< 0.05$ | $< 0.001$ | $< 0.05$ | $< 0.001$ | $< 0.05$ | $< 0.001$ | $< 0.001$ |

flect different value control cases. (1) Two-dimensional target: we select any two value dimensions and assign them priority weights of 1, while setting the weight of others to 0. (2) Random-dimensional target: we assign a random priority weight in the interval $[0, 1]$ to each value dimension independently. (3) Full-dimensional target: all value dimensions are required to be fully aligned, with a weight of 1. Given a test sample and a value objective, we first determine the right answer as the candidate answer whose annotated value profile is closest to the specified value objective. Alignment performance is then measured as the accuracy that the model selects the right answer. Furthermore, we conduct human evaluation to verify the consistency among human judgments, model selections, generate text and the assigned labels. In addition, for text generation, we evaluate the

outputs using both an LLM-based judge and human annotators, achieving an 85% consistency between evaluation accuracy and the dataset labels. For reproducibility, detailed evaluation metrics are provided in the appendix D.3.

**Baselines** We compare DisAlign with representative pluralistic alignment baselines spanning three groups: (i) *Inference-time or Prompting-based Methods*, including **Steerable Pluralism** (Adams et al., 2025), **Modular Pluralism** (Feng et al., 2024) and a naive prompting baseline **Prompt-Align** implemented by us; (ii) *Fine-tuning based Methods*, including **MAP** (Wang et al., 2024b) and **CPO** (Guo et al., 2024). (iii) *Model-merging based Methods*, including **PAS** (Zhu et al., 2024), **Pmol** (Liu et al., 2024), **Rewarded Soups** (Rame et al., 2023) and variants with

advanced model-merging techniques like **FWA** (Matena & Raffel, 2022), **DARE** (Yu et al., 2024), **AlignMerge** (Roy et al., 2025), **KnOTS** (Stoica et al., 2025), **VISPA** (Zheng et al., 2026). Full baseline details are in Appendix D.2.

**Implementation Details** We instantiate our method on Llama-3.2-3B (Grattafiori et al., 2024) and Qwen3.5-4B (Bazi et al., 2026), following established protocols (Matena & Raffel, 2022; Yadav et al., 2023; Stoica et al., 2025) to train value-specific experts using full-parameter SFT (Ouyang et al., 2022). To ensure fairness and reproducibility, all experts are initialized from the same pretrained checkpoint, trained with identical hyperparameters, and early-stopped on a disjoint validation set. Training and evaluation data are strictly separated to prevent leakage, ensuring that observed parameter differences arise solely from value specific objectives. All experiments are conducted on $3\times$ NVIDIA A800 (60GB) GPUs. All experiments reported in the main text are conducted under the stated assumptions, and experiments validating these assumptions are provided in the appendix A. Full details for implementation are provided in Appendix D.3.

**Research Questions** In next subsections, we conduct extensive experiments to explore two main research questions:
• **RQ1**. Does our decomposition framework DisAlign enable more precise value weight control compared to baselines in pluralistic value alignment?
• **RQ2**. How do different components in DisAlign contribute to the improvement?

## 4.2. Performance on Pluralistic Value Alignment (RQ1)

**Main Results** We first evaluate whether DisAlign can achieve pluralistic alignment in multi-dimensional value alignment tasks. As shown in Table 1, DisAlign consistently outperforms all baseline methods in alignment accuracy under all three target-setting regimes. In half of the evaluation metrics, its performance exceeds the second-best method by more than three standard deviations. In addition, under the *Random-dimensional* setting, changing the alignment weight proportions leads to much larger performance gains than for other methods, demonstrating that DisAlign supports fine-grained and controllable alignment. Figure 2 further shows that DisAlign has minimal impact on non-target value dimensions, indicating stable alignment with strong resistance to interference. In contrast, other methods achieve only partial multi-dimensional alignment but introduce significant shifts in non-target value dimensions. Overall, these results show that DisAlign achieves accurate alignment on target values while effectively isolating non-target value dimensions. Full results included Mistral-7B are provided in Appendix E.

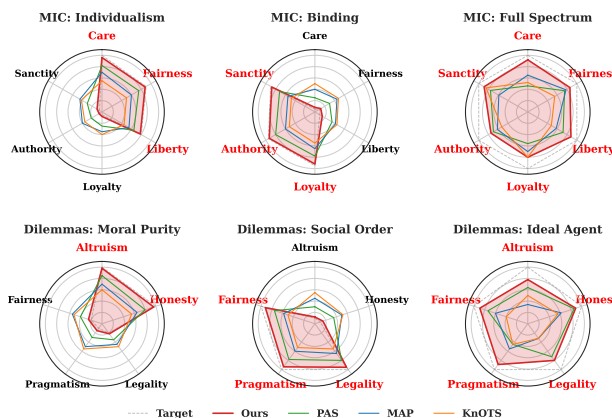

*Figure 2.* Alignment performance under different numbers of value dimensions to be aligned on the MIC and Dilemmas datasets.

**Fine-grained Value Priority Control.** We evaluate DisAlign under different value priority settings from two aspects. First, we conduct a *Full inversion experiment* to test whether the model can switch between two conflicting values (Care vs. Loyalty). Second, we run a *fine-grained sensitivity experiment* to assess how the model responds to small weight changes in multi-value alignment. In our experiments, we explicitly remove the consensus component and operate in value specific subspaces. As shown in Table 2, the model consistently keeps low scores on non-target values and shows clear and significant alignment changes before and after weight adjustment. These results indicate that our method achieves precise control in fine-grained settings and avoids interference from other values in single-value alignment. The complete results for both experiments are provided in the Appendix E.1.

## 4.3. Ablation Study (RQ2)

**Function of Consensus and value specific Components** On the MIC dataset, we first build a consensus set $\mathcal{D}_\cap$ using samples that express both Truth and Fairness. We then form a new set, special parts $\mathcal{D}_\Delta$ by removing these overlapping samples. Using MMLU (Hendrycks et al., 2020), ARC-C (Clark et al., 2018), and related benchmarks, we evaluate models under different component combinations.

As shown In Table 3, the results illustrate the specific role of each component. The consensus components preserves general capabilities and shared values but performs poorly on value specific preferences. In contrast, the value specific subspace effectively resolves value Special parts with a small drop in general performance. Combining the consensus component with the value specific subspace preserves overall performance while enabling precise value alignment. These results confirm that DisAlign enables precise value control without harming general capabilities, making it an effective approach for pluralistic alignment. Broader evaluation results are reported in Appendix E.3.

*Table 2.* Results of the **Full Inversion Experiment** (Care-Loyalty Switch) and the **Fine-grained Sensitivity Experiment** (Contradictory Zone Weight Adjustment). We report the performance for both objectives following the weight adjustment, the magnitude of the response offset ($\Delta$), and the control fidelity $\phi = \mathrm{CosSim}(\Delta\mathbf{w}, \Delta\mathbf{P})$, where $\Delta\mathbf{w}$ and $\Delta\mathbf{P}$ denote the changes in control weights and model scores, respectively. Only Fine-grained Sensitivity the $\beta = 1$.

| Method | Full inversion experiment $(w_A, w_B) : (0.01, 1.0) \rightarrow (1.0, 0.01)$ | | | Fine-grained sensitivity experiment $(w_A, w_B) : (0.5, 0.7) \rightarrow (0.7, 0.5)$ | | | |
|---|---|---|---|---|---|---|---|
| | $A_{after}$ ($\uparrow$) | $B_{after}$ ($\downarrow$) | **Swap Success** | $\Delta A$ (Gain) ($\uparrow$) | $\Delta B$ (Suppression) ($\uparrow$) | **Fidelity** ($\phi$) | **State** |
| KnOTS | 55.6 | 44.8 | Partial | -0.4 (stagnant) | -0.2 | 0.12 | Rigid |
| Rew. Soups | 56.1 | 43.3 | Partial | +0.1 (insignificant) | +0.3 | 0.05 | Rigid |
| MAP | **56.2** | 42.9 | ✓ | +1.6 | -0.2 | 0.78 | Responsive |
| **DisAlign** | 55.3 | **39.9** | ✓✓ (Cleanest) | **+1.2** | **-2.3** | **0.92** | **Precise** |

*Table 3.* Ablation study results on both general capabilities and shared/Special parts alignment datasets.(llama3.2-3B)

| Config. | General Capabilities | | Shared | Special parts |
|---|---|---|---|---|
| | MMLU | ARC-C | $\mathcal{D}_{\cap}$ | $\mathcal{D}_{\Delta}$ |
| $\theta_0$ | **63.1** | **78.1** | 67.1 | 30.7 |
| $\theta_c^* + \beta\mathcal{C}_r$ | 62.8 | 76.5 | **94.2** | 30.5 |
| $\theta_0 + \alpha U_v$ | 46.5 | 50.2 | 62.1 | **54.4** |
| $\theta_c^* + \beta\mathcal{C}_r + \alpha U_v$ | 62.1 | 77.2 | **95.0** | 53.5 |

*Table 4.* We report the alignment accuracy under Random-dimensional (B) and Full-dimensional (C) settings, formatted as *B. Random / C. Full*, alongside the computational time cost (s) on MIC dataset. **CR** denotes the Consensus Removal operation, and **Decomp.** specifies the geometric metric used for value decomposition.

| Method | Geometry | Decomp. | CR | Complexity | Time | MIC | Daily D. |
|---|---|---|---|---|---|---|---|
| Average | – | – | – | $\mathcal{O}(1)$ | 1 | 40.2 / 48.7 | 39.6 / 48.6 |
| FWA | Fisher | Weighted | – | High | ~270 | 46.3 / 50.1 | 49.1 / 52.3 |
| Euclid-$I$ | Euclidean | $\ell_2$ Norm | No | $\mathcal{O}(d)$ | ~120 | 40.8 / 48.9 | 39.6 / 48.6 |
| Euclid-$I$ | Euclidean | $\ell_2$ Norm | Yes | $\mathcal{O}(d)$ | ~110 | 46.9 / 50.8 | 49.7 / 52.8 |
| Diag Fisher | Fisher | Diagonal | No | $\mathcal{O}(d)$ | ~320 | 45.8 / 49.7 | 48.3 / 51.2 |
| Diag Fisher | Fisher | Diagonal | Yes | $\mathcal{O}(d)$ | ~315 | 58.3 / 55.5 | 59.2 / 56.8 |
| **Ours** | **Fisher** | **Subspace** | **Yes** | $\mathcal{O}(m^3)$ | **260** | **59.6 / 56.6** | **60.6 / 57.1** |

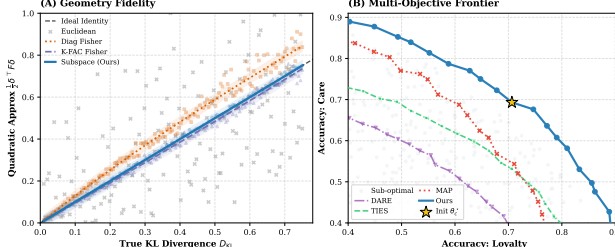

*Figure 3.* **(A)** Metric fidelity These represent the changes in weights and the corresponding changes . **(B)** Pareto frontier illustrating Fisher-guided steering under conflicting objectives.

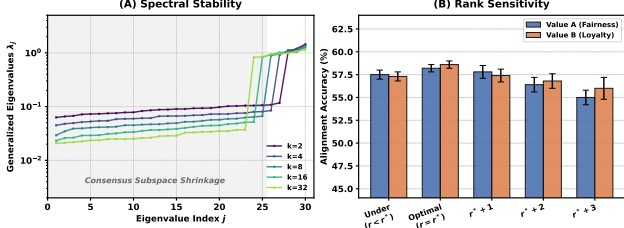

*Figure 4.* **(A)** GEVP feature spectra under different settings. **(B)** Ablation results with varying consensus rank $r$.

**Information-Geometric Space Analysis**   To demonstrate the advantage of the information-geometric space, we compare it with Euclidean-space methods in approximating KL divergence and in Pareto frontier analysis. As shown in Figure 3A, the subspace Fisher distance exhibits an approximately linear relationship with local KL divergence and shows a much stronger correlation than Euclidean distance. This result indicates that consensus defined in the information-geometric space better reflects probabilistic consistency in model behavior. Furthermore, Figure 3B shows that in multi-objective control, our method achieves a superior Pareto frontier without sacrificing other objectives. Together, these results confirm both the theoretical validity and practical effectiveness of information-geometric guidance for value control.

**Geometry and Efficiency**   Finally, we examine the roles of the consensus component and the value-specific subspace under different geometric settings. We estimate the em-

pirical Fisher Information Matrix (FIM) using $N$=5120 samples and adopt a low-rank Fisher approximation, which enables scalable $\mathcal{O}(m^3)$ matrix inversion with efficient storage. As shown in Table 4, the ablation results indicate that both consensus removal (CR) and decomposition in Fisher geometry are essential for robust and controllable value modulation. When CR is omitted, performance drops to the level of standard information-geometric aggregation methods (e.g., FWA). Moreover, although applying CR in Euclidean space yields modest improvements, Euclidean-based decomposition remains insufficient to achieve stable and effective multi-value alignment.

### 4.4. Sensitivity Analysis

**Sensitivity of Consensus Components and Value-Specific Subspaces**   We perform sensitivity analyses from three as-

*Table 5.* **Scalability Analysis of Value Subspaces with Increasing Expert Count** ($k$). We monitor the evolution of the effective value-specific subspace dimensionality ($m$) and the average pairwise correlation among value directions across varying scales of $k$.

| Expert Count $k$ | value Subspace Size $m$ | Avg. Correlation |
|---|---|---|
| **Low** ($2 \sim 5$) | $m = k$ | $\approx 0.02$ |
| **Medium** ($5 \sim 15$) | $k - 2 < m \leq k$ | $\approx 0.05$ |
| **High** ($\geq 15$) | $k - 4 \leq m \leq k - 2$ | $\approx 0.15$ |

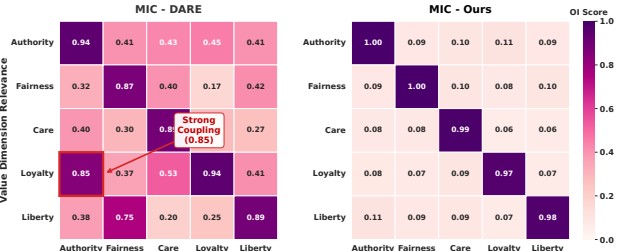

*Figure 5.* We analyze the pairwise cosine correlations of value representations on the MIC dataset. Comparing the settings with the consensus component **retained** versus **removed**.

pects to assess the robustness of our model on MIC datasets. First, as shown in Figure 5, retaining the consensus component leads to strong correlations across value dimensions, while explicitly removing it substantially reduces inter-value correlations and enables more precise value control. Second, Figure 4 shows that the consensus subspace is geometrically stable: a clear spectral gap consistently appears across different value scales, and model performance remains stable when the rank $r$ is underestimated, with only minor degradation under overestimation due to value leakage. Third, Table 5 analyzes scalability by examining how the size of value-specific subspaces and the average inter-value correlation evolve as the number of aligned values increases. Taken together, these results demonstrate the stability of our spectral truncation strategy, the weak coupling among value-specific subspaces, and confirm that consensus removal is essential for reliable value disentanglement. We have included additional stability analysis of the spectral gap across more models and datasets in the Appendix E.6.

**Open-ended Generation Evaluation.** We further evaluate the open-ended generation capabilities of our method on the MIC dataset using Llama-3.2-3B. To rigorously assess the outputs, we employ GPT-4o as an evaluator to score each response along two axes: **Value Alignment** (rated 1–5, measuring how well the output reflects the target value combination) and **Response Quality** (rated 1–5, measuring fluency and coherence).

As shown in Table 6, DisAlign achieves significantly better value alignment in both random-dimensional and two-dimensional targets compared to baseline methods (FWA and DARE), while maintaining highly comparable response quality.

*Table 6.* Open-ended generation evaluation on the MIC dataset. GPT-4o evaluates the responses on Value Alignment (1–5) and Response Quality (1–5). **Bold** indicates the best performance.

| Method | Value Align. ↑ | | Response Quality ↑ | Win Rate vs. Ours ↓ |
|---|---|---|---|---|
| | Random | Two-dim | | |
| FWA | 2.4 | 2.9 | 3.4 | 27% |
| DARE | 2.8 | 3.1 | **3.8** | 31% |
| **DisAlign (Ours)** | **4.2** | **4.4** | 3.7 | – |

Furthermore, Table 7 provides qualitative examples of our method's generations under various value constraints. Given the same prompt, the generated responses seamlessly adapt to single values (e.g., Care or Loyalty) as well as complex blended targets (e.g., 60% Care + 40% Loyalty). These quantitative and qualitative results together demonstrate that our method is highly effective in open-ended generation tasks.

*Table 7.* Qualitative examples of open-ended generation by DisAlign under different value targets.

| **Prompt:** *"My friend asked me to invest in their risky business."* | |
|---|---|
| **Target Value** | **Generated Response** |
| Care | "You should be honest with your friend about the risks." |
| Loyalty | "You should consider supporting your friend's business." |
| 60% Care + 40% Loyalty | "You should talk to your friend about the risks involved, but also show that you care about the relationship." |

## 5. Conclusion

We present a model fisher merge framework for pluralistic alignment that addresses the structural tension between shared model capabilities and diverse value specifications. By formulating alignment as a representation decomposition problem within an information geometric framework, our method explicitly separates a stable *consensus subspace*, capturing shared semantics and general capabilities, from Fisher orthogonal residual directions that encode value specific preferences. Both theory and experiments demonstrate the effectiveness and stability of our method.

## Acknowledgements

This work is supported by National Natural Science Foundation of China (NSFC) under the Grant No. 62372113, and Major Project of the National Social Science Foundation of China (NSSFC) under the Grant No. 25&ZD260. Peng Zhang & Tun Lu are with the College of Computer Science and Artificial Intelligence, Fudan University. Tun Lu is also affiliated to the Silver-X MOE Philosophy & Social Sciences Laboratory, Fudan Institute on Aging, and Shanghai Key Laboratory of Data Science, Fudan University.

## Impact Statement

This paper presents a geometric framework for pluralistic value alignment. Our experiments rely entirely on standard, open-source benchmarks and do not involve human trials or sensitive personal information. While the subject matter concerns value representation, our goal is to enhance the controllability and safety of LLMs. We do not foresee any direct negative ethical consequences resulting from the research process or the experimental methodology.

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

# A. Assumption Details

## A.1. Empirical Validation of Theoretical Assumptions

The analytical tractability of our proposed Fisher-weighted consensus geometry relies on the premise that the value-alignment process occurs within a stable region of the policy's information manifold. We formalize this requirement via the following regularity condition:

**Assumption A.1** (Local Quadratic Regime and Smoothness). Let $\mathcal{P} = \{p_\theta : \theta \in \Theta\}$ be the statistical manifold of the policy. We assume that all value-aligned experts $\{\theta_\mathcal{A}^{(i)}\}_{i=1}^k$ reside within a sufficiently small $\delta$-neighborhood of the base policy $\theta_0$, such that $\|\theta_\mathcal{A}^{(i)} - \theta_0\|_2 \leq \delta$. Within this neighborhood, the following properties hold:

1. **Regularity:** The KL divergence $D_{\mathrm{KL}}(p_{\theta_0}\|p_\theta)$ is a $C^2$ smooth function with respect to $\theta$.

2. **Local Curvature:** The KL divergence admits a valid second-order Taylor approximation at $\theta_0$:

$$D_{\mathrm{KL}}(p_{\theta_0}\|p_\theta) \approx \frac{1}{2}(\theta - \theta_0)^\top F(\theta_0)(\theta - \theta_0), \quad (10)$$

where $F(\theta_0)$ denotes the Fisher Information Matrix (FIM) of the base policy.

If the higher-order residuals of this expansion were dominant, the analytical solution derived in Eq. equation 3 would deviate significantly from the true $I$-projection. We provide comprehensive empirical evidence in Figure 6 to justify the validity of this quadratic regime.

**Local Validity along the Interpolation Path.** We first examine the KL divergence along the geodesic-proximal path $\theta(\alpha) = \theta_0 + \alpha(\theta_\mathcal{A}^{(i)} - \theta_0)$ for $\alpha \in [0, 1.5]$. As illustrated in Figure 6A, the empirically measured $D_{\mathrm{KL}}$ (blue dots) exhibits a near-perfect quadratic trajectory that aligns closely with our Fisher-based surrogate (red dashed line). Crucially, the independently aligned experts (represented at $\alpha = 1.0$) reside well within the identified *Quadratic Trust Region*. In this region, the relative approximation error remains strictly below 4%, confirming that the second-order term captures the vast majority of the information-theoretic displacement during alignment.

**Geometric Consistency in the parameter space.** To ensure that this stability is not restricted to a specific alignment direction, we evaluate the approximation across the entire parameter space. Figure 6B presents a parity plot comparing the Fisher-predicted distances against measured KL divergences for 400 random perturbations within $\mathcal{S}$. The high Spearman correlation ($\rho > 0.99$) and minimal heteroscedastic drift from the identity line demonstrate that the FIM provides a high-fidelity, direction-invariant characterization of the local value geometry.

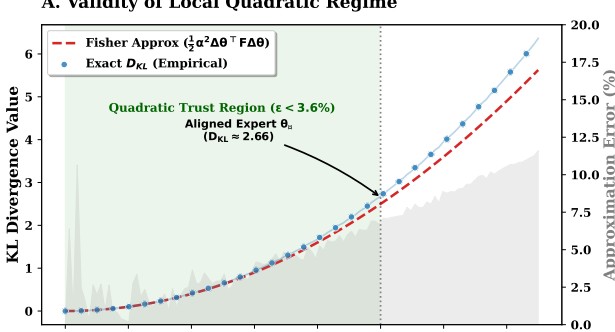

**A. Validity of Local Quadratic Regime**

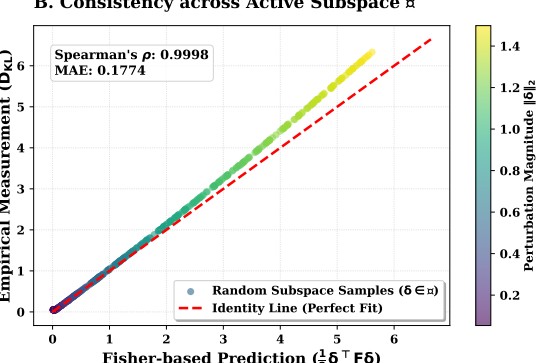

**B. Consistency across Active Subspace ¤**

*Figure 6.* **Empirical validation of the Local Quadratic Regime (Assumption A.1). (A) Local validity along the interpolation path:** The Fisher-based quadratic approximation (red dashed line) matches the empirically measured KL divergence (blue dots) with high fidelity. The green-shaded region indicates the *Quadratic Trust Region* where the relative approximation error remains below 4%, effectively covering the independently aligned expert $\theta_\mathcal{A}$ ($\alpha = 1.0$). **(B) Subspace consistency:** Parity plot comparing predicted vs. measured $D_{\mathrm{KL}}$ for 400 random samples within the parameter space. The high Spearman correlation ($\rho > 0.99$) and minimal drift from the identity line demonstrate that the Fisher Information Matrix serves as a reliable metric for the local value geometry.

These results provide a solid empirical foundation for our method: by confirming that alignment experts remain within the trust region of $\theta_0$, we justify replacing the computationally intractable $I$-projection with the efficient, Fisher-weighted closed-form solution in Proposition 3.3.

**Mitigation Strategy: Iterative Refinement via Trust-Region Updates.** To address potential violations of the Local Quadratic Regime (Assumption A.1) under large distributional shifts (e.g., extensive RLHF), we relax the one-shot projection requirement. Instead, we adopt a damped iterative scheme that ensures the consensus optimization trajectory remains strictly within the valid information-geometric neighborhood.

Let $\theta^{(t)}$ denote the consensus estimate at iteration $t$ (initialized at $\theta^{(0)} = \theta_0$). We compute the optimal update direction $\mathbf{d}^{(t)}$ effectively as a Newton step on the aggregate quadratic

surface:

$$\mathbf{d}^{(t)} = \left( \sum_{i=1}^{k} \tilde{F}_i \right)^{-1} \sum_{i=1}^{k} \tilde{F}_i \left( \theta_{\mathcal{A}}^{(i)} - \theta^{(t)} \right). \quad (11)$$

Crucially, rather than applying the full update, we introduce a step size $\eta_t \in (0, 1]$ determined via a backtracking line search. The step size is selected to satisfy the *Trust Region Condition*:

$$\frac{\left| D_{\text{KL}}(p_{\theta^{(t)}} \| p_{\theta^{(t)} + \eta_t \mathbf{d}^{(t)}}) - \hat{Q}(\eta_t) \right|}{D_{\text{KL}}(p_{\theta^{(t)}} \| p_{\theta^{(t)} + \eta_t \mathbf{d}^{(t)}})} \leq \epsilon_{\text{tol}}, \quad (12)$$

where $\hat{Q}(\eta_t) = \frac{1}{2} \eta_t^2 (\mathbf{d}^{(t)})^\top \bar{F} \mathbf{d}^{(t)}$ is the local quadratic prediction and $\epsilon_{\text{tol}}$ is the empirical tolerance threshold (e.g., 0.07 as established in Sec. A.1). The parameter update $\theta^{(t+1)} \leftarrow \theta^{(t)} + \eta_t \mathbf{d}^{(t)}$ is performed only when this local metric fidelity is guaranteed. This procedure decomposes a potentially large, geometry-violating shift into a sequence of smaller, valid geodesic steps, thereby preserving the theoretical guarantees of the Fisher-weighted consensus even in regimes of significant model adaptation.

**Stress Test: Robustness under Assumption Violation**
*The solution used in the main text is a single-step solution, not the pressure scenario described below.* Experimental Setup. To rigorously evaluate the limits of our theoretical framework, we designed a "Stress Test" scenario that intentionally violates the *Local Quadratic Regime* (Assumption A.1). We trained a specialized expert model $\theta_{\text{stress}}$ using an aggressive RLHF configuration , resulting in a policy that is "over-optimized" for a specific reward function. This induces a significant distributional shift, pushing the expert parameters far beyond the trust region of the base policy $\theta_0$. Our goal is to quantify the breakdown of the one-shot Fisher projection and demonstrate the restorative capability of the proposed Iterative Trust-Region update.Results: Divergence and Recovery.The results are visualized in Figure 7.First, we observe that for the over-optimized expert, the actual KL divergence manifold (blue solid line) deviates sharply from the quadratic approximation predicted by the base model's Fisher matrix (red dashed line).At the expert's position ($\alpha \approx 2.8$), the *approximation gap*—the discrepancy between the predicted and actual information geometry—exceeds $300\%$.Consequently, applying the standard closed-form solution (One-shot Projection) results in a consensus policy that overshoots the low-loss basin, degrading the KL-control capability.In contrast, the Iterative Trust-Region method (green markers) successfully navigates this non-quadratic landscape. By decomposing the large shift into a sequence of smaller updates validated against the trust-region condition (Eq. 12), the algorithm effectively re-calibrates the local metric at each step. As shown by the green trajectory, the iterative updates closely track the true

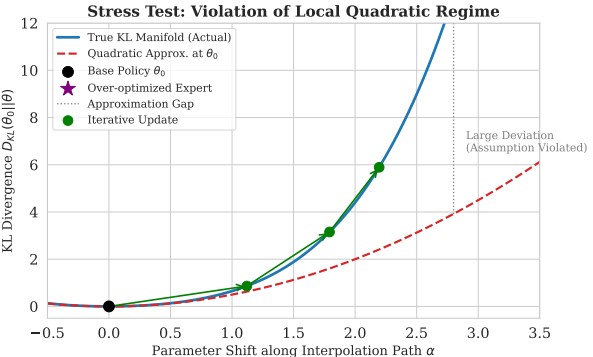

*Figure 7.* Stress Test under Distributional Shift. The plot illustrates the divergence between the true information geometry (Blue) and the local quadratic approximation (Red) when an expert is over-optimized (far from $\theta_0$). The standard one-shot approach fails due to the large approximation gap. However, our Iterative Update strategy (Green path) recovers performance by taking strictly validated steps along the manifold.

manifold, converging to a stable consensus even when the initial static assumption $\bar{F}$ is no longer globally valid.This confirms that while the local quadratic assumption is necessary for analytical tractability, our iterative algorithm provides a robust safeguard against practical violations caused by strong model adaptation.

### A.2. Empirical Justification of Local Metric Homogeneity

We give the assumption 3.2 restated:

**Assumption A.2** (Local Metric Homogeneity). Let $\bar{F} = \frac{1}{k} \sum_{i=1}^{k} \tilde{F}_i$ denote the average empirical Fisher Information Matrix restricted to the parameter space. We assume that:

1. The representative metric $\bar{F}$ is symmetric positive definite ($\bar{F} \succ 0$) and possesses a spectral profile consistent with the population;

2. The individual Fisher matrices $\{\tilde{F}_i\}_{i=1}^{k}$ exhibit bounded deviation from $\bar{F}$ and share a unified topological structure in the neighborhood of the anchor $\theta_c^\star$.

The analytical validity of our PoE-induced consensus relies on Assumption A.2, which posits that diverse value experts share a consistent local geometry within the parameter space. We provide a multi-dimensional empirical validation based on the results in Figure 8.

**Structural Alignment via CKA.** To quantify the geometric similarity between experts, we employ the Centered Kernel Alignment (CKA) index. As shown in Figure 8 (A1), the pairwise CKA scores across different experts consistently cluster around the $0.60$ range (e.g., $0.58 \sim 0.62$).

While distinct from identity (1.0), this uniform moderate alignment indicates a significant *structural consistency*. It suggests that although experts are specialized for disjoint values (e.g., Authority vs. Liberty), their sensitivity landscapes remain anchored to a shared manifold derived from the base model, preventing geometric orthogonality.

**Spectral Consistency.** We further investigate the curvature information by analyzing the eigen-spectra of the Fisher matrices. Figure 8 (A2) reveals that the spectra of all individual experts follow a nearly identical power-law decay pattern ($\lambda_j \propto j^{-\beta}$). Notably, the representative metric $\bar{F}$ (red dashed line) faithfully preserves this spectral envelope. This synchronization in eigenvalue decay confirms that all experts share the same "information dimensionality" and curvature complexity, validating $\bar{F}$ as a topologically faithful representative of the population geometry.

**Bounded Deviation and Convergence.** We evaluate the approximation fidelity by measuring the relative Frobenius distance $\|\tilde{F}_i - \bar{F}\|_F / \|\bar{F}\|_F$. Figure 8 (B1) demonstrates that the relative error is strictly bounded within a narrow interval ($0.71 \sim 0.77$). The remarkably small inter-quartile range implies that no single expert acts as a geometric outlier; rather, all experts are equidistantly distributed around the consensus mean. Furthermore, Figure 8 (B2) illustrates the scalability of our approach: as the number of experts $k$ increases, the estimation error of the global geometry decreases monotonically (dropping from $0.85$ to $0.37$). This convergence property ($\approx \mathcal{O}(1/\sqrt{k})$) empirically justifies our use of the average Fisher matrix $\bar{F}$ as a robust consensus anchor.

## B. Proof of Proposition 3.3

This appendix provides a fully rigorous derivation of Proposition 3.3, which characterizes the optimal *consensus subspace projection* of independently aligned experts under a local information-geometric approximation.

Throughout this section, we assume that Assumption A.1 (Local Quadratic Regime and Smoothness) and Assumption A.2 hold. All asymptotic equivalences are understood in the limit $\|\theta - \theta_0\|_2 \to 0$.

We proceed in two steps:

1. Lemma B.1 establishes that, up to second order, the $I$-projection of the Product-of-Experts (PoE) distribution is equivalent to minimizing a Fisher-weighted quadratic objective.

2. Lemma B.2 derives the unique closed-form minimizer when this objective is restricted to a low-rank consensus subspace.

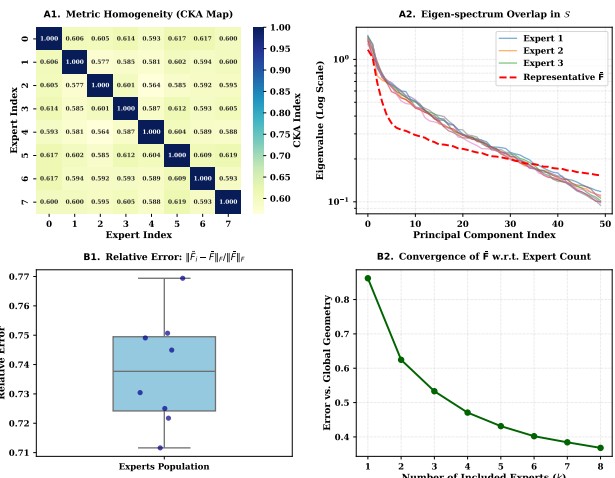

*Figure 8.* **Empirical validation of Local Metric Homogeneity.** **(A1) Metric Similarity:** Pairwise CKA scores ($\approx 0.60$) indicate a stable structural alignment across experts. **(A2) Spectral Consistency:** The synchronized power-law decay of eigen-spectra confirms that experts share consistent curvature properties with the representative mean $\bar{F}$ (dashed line). **(B1) Bounded Deviation:** The narrow spread of relative errors shows that experts are uniformly distributed around the consensus geometry. **(B2) Convergence:** The monotonic decrease in approximation error with increasing expert count $k$ validates the robustness of the consensus estimation.

**Lemma B.1** (Quadratic Approximation of PoE $I$-Projection). *Let*

$$\tilde{p}_{\text{poe}}(\tau) \propto \prod_{i=1}^{k} p_{\theta_{\mathcal{A}}^{(i)}}(\tau)$$

*denote the Product-of-Experts distribution induced by independently aligned experts. Under Assumption 3.1, minimizing*

$$D_{\text{KL}}(\tilde{p}_{\text{poe}} \parallel p_\theta)$$

*with respect to $\theta$ is locally equivalent, up to second order in $\|\theta - \theta_0\|_2$, to minimizing the quadratic objective*

$$\mathcal{L}(\theta) = \sum_{i=1}^{k} (\theta - \theta_{\mathcal{A}}^{(i)})^\top F_i (\theta - \theta_{\mathcal{A}}^{(i)}), \tag{13}$$

*where $F_i$ denotes the Fisher Information Matrix of expert $i$, evaluated in a neighborhood of the base policy $\theta_0$.*

*Proof.* Instead of relying on the ambiguous decomposition of the PoE expectation, we adopt a rigorous information-geometric perspective. We formulate the consensus anchor $\theta^*$ as the geometric barycenter that minimizes the cumulative information loss (KL divergence) from all expert distributions $\{p_{\theta_{\mathcal{A}}^{(i)}}\}_{i=1}^{k}$ to the merged model $p_\theta$:

$$\theta^* = \arg\min_\theta \sum_{i=1}^{k} D_{\text{KL}}\left(p_{\theta_{\mathcal{A}}^{(i)}} \parallel p_\theta\right).$$

Consider a single term $D_{\mathrm{KL}}(p_{\theta_{\mathcal{A}}^{(i)}} \| p_\theta)$. Under Assumption 3.1, we perform a second-order Taylor expansion with respect to $\theta$ around the expert parameter $\theta_{\mathcal{A}}^{(i)}$:

$$
\begin{aligned}
D_{\mathrm{KL}}(p_{\theta_{\mathcal{A}}^{(i)}} \| p_\theta) &= D_{\mathrm{KL}}(p_{\theta_{\mathcal{A}}^{(i)}} \| p_{\theta_{\mathcal{A}}^{(i)}}) \\
&+ (\theta - \theta_{\mathcal{A}}^{(i)})^\top \nabla_\theta D_{\mathrm{KL}}(p_{\theta_{\mathcal{A}}^{(i)}} \| p_\theta)\big|_{\theta = \theta_{\mathcal{A}}^{(i)}} \\
&+ \frac{1}{2}(\theta - \theta_{\mathcal{A}}^{(i)})^\top H_i (\theta - \theta_{\mathcal{A}}^{(i)}) + O(\|\theta - \theta_{\mathcal{A}}^{(i)}\|^3).
\end{aligned}
\tag{14}
$$

We invoke standard properties of the KL divergence for well-specified parametric models:

1. **Zeroth-order term:** The divergence of a distribution from itself is zero, so $D_{\mathrm{KL}}(p_{\theta_{\mathcal{A}}^{(i)}} \| p_{\theta_{\mathcal{A}}^{(i)}}) = 0$.

2. **First-order term:** The gradient vanishes at the minimum (where $\theta = \theta_{\mathcal{A}}^{(i)}$), i.e., $\nabla_\theta D_{\mathrm{KL}}(p_{\theta_{\mathcal{A}}^{(i)}} \| p_\theta)|_{\theta_{\mathcal{A}}^{(i)}} = 0$.

3. **Second-order term:** The Hessian matrix of the KL divergence $D_{\mathrm{KL}}(p_{\theta^*} \| p_\theta)$ with respect to $\theta$ evaluated at $\theta^*$ is exactly the Fisher Information Matrix (FIM). Thus, $H_i = F_i$.

Substituting these terms back into the objective function and summing over all $k$ experts, the optimization problem simplifies to:

$$
\mathcal{L}(\theta) \approx \sum_{i=1}^{k} \frac{1}{2}(\theta - \theta_{\mathcal{A}}^{(i)})^\top F_i (\theta - \theta_{\mathcal{A}}^{(i)}).
$$

This confirms that under the local quadratic regime, minimizing the sum of KL divergences is mathematically equivalent to minimizing the Fisher-weighted quadratic objective in Eq. equation 13. This derivation provides a robust geometric justification for the consensus anchor without requiring the additivity of log-densities under the PoE expectation. $\square$

**Lemma B.2** (Optimal Consensus within a Subspace). *Let $\mathcal{S} = \theta_0 + \mathrm{span}(Q)$ be an $m$-dimensional consensus subspace, where $Q \in \mathbb{R}^{d \times m}$ has orthonormal columns. Assume that $\sum_{i=1}^{k} \tilde{F}_i \succ 0$, where $\tilde{F}_i = Q^\top F_i Q$. Then the unique minimizer of $\mathcal{L}(\theta)$ over $\theta \in \mathcal{S}$ is given by*

$$
\theta_c^\star = \theta_0 + Q \left( \sum_{i=1}^{k} \tilde{F}_i \right)^{-1} \sum_{i=1}^{k} \tilde{F}_i Q^\top (\theta_{\mathcal{A}}^{(i)} - \theta_0). \tag{15}
$$

*Proof.* Any $\theta \in \mathcal{S}$ can be written uniquely as $\theta = \theta_0 + Q\tilde{\theta}$ for some $\tilde{\theta} \in \mathbb{R}^m$. Substituting this parameterization into

Eq. equation 13 gives

$$
\mathcal{L}(\tilde{\theta}) = \sum_{i=1}^{k} (Q\tilde{\theta} - \Delta\theta^{(i)})^\top F_i (Q\tilde{\theta} - \Delta\theta^{(i)}),
$$

where $\Delta\theta^{(i)} = \theta_{\mathcal{A}}^{(i)} - \theta_0$.

Expanding and differentiating with respect to $\tilde{\theta}$ yields

$$
\nabla_{\tilde{\theta}} \mathcal{L} = 2 \sum_{i=1}^{k} Q^\top F_i (Q\tilde{\theta} - \Delta\theta^{(i)}).
$$

Setting the gradient to zero gives the normal equations

$$
\left( \sum_{i=1}^{k} Q^\top F_i Q \right) \tilde{\theta} = \sum_{i=1}^{k} Q^\top F_i \Delta\theta^{(i)}.
$$

Defining $\tilde{F}_i = Q^\top F_i Q$ and $\tilde{\delta}^{(i)} = Q^\top \Delta\theta^{(i)}$, we obtain

$$
\left( \sum_{i=1}^{k} \tilde{F}_i \right) \tilde{\theta} = \sum_{i=1}^{k} \tilde{F}_i \tilde{\delta}^{(i)}.
$$

Since $\sum_{i=1}^{k} \tilde{F}_i \succ 0$ by assumption, the solution is unique and given by

$$
\tilde{\theta}^\star = \left( \sum_{i=1}^{k} \tilde{F}_i \right)^{-1} \sum_{i=1}^{k} \tilde{F}_i \tilde{\delta}^{(i)}.
$$

Substituting back into $\theta = \theta_0 + Q\tilde{\theta}$ yields Eq. equation 15. $\square$

**Proof of Proposition 3.3** Lemma B.1 shows that, within the local quadratic regime, the PoE $I$-projection reduces to minimizing $\mathcal{L}(\theta)$. Lemma B.2 characterizes the unique minimizer of this objective over the consensus subspace $\mathcal{S}$. Combining the two results yields $\theta_c^\star = \theta_0 + Q\tilde{\theta}^\star$, which proves Proposition 3.3. $\square$

## C. Proof of Proposition C.1

We restate the proposition for completeness. The Proposition 3.4 (Restated)

**Proposition C.1** ( Stability and Identifiability of the Consensus Subspace). *Let $\theta_{\mathbf{c}}^\star$ denote the PoE anchor and $\mathbf{r}_i = (\theta_{\mathcal{A}}^{(i)} - \theta_{\mathbf{c}}^\star)$ the projected residuals in the parameter space. Consider the Generalized Eigenvalue Problem (GEVP) defined in Eq. equation 5:*

$$
\Sigma_r \mathbf{b}_j = \lambda_j \tilde{F} \mathbf{b}_j.
$$

*Let $\mathcal{C}_r = \mathrm{span}\{\mathbf{b}_1, \ldots, \mathbf{b}_r\}$ be the subspace associated with the $r$ smallest generalized eigenvalues. Under Assumption A.2:*

1. **(Stability)** *If a spectral gap exists such that $\lambda_r < \lambda_{r+1}$, the subspace $\mathcal{C}_r$ is unique up to an orthogonal transformation. Furthermore, for any symmetric perturbation $\Delta$ to the residual covariance $\Sigma_r$ with $\|\Delta\|_2 \leq \epsilon$, the perturbed subspace $\widehat{\mathcal{C}}_r$ satisfies:*

$$d(\mathcal{C}_r, \widehat{\mathcal{C}}_r) \leq O\left(\frac{\kappa(\tilde{F}) \cdot \epsilon}{\lambda_{r+1} - \lambda_r}\right),$$

*where $d(\cdot, \cdot)$ denotes the sine of the largest principal angle, and $\kappa(\tilde{F})$ is the condition number of the aggregate Fisher metric.*

2. **(Identifiability / Translation Invariance)** *For any global shift $v \in \mathbb{R}^d$, define the shifted experts $\theta_{\mathcal{A}}^{(i)\prime} = \theta_{\mathcal{A}}^{(i)} + v$. Under the local quadratic approximation of the PoE objective, the consensus geometry is invariant, i.e., the induced consensus subspace $\mathcal{C}'_r$ coincides with $\mathcal{C}_r$ up to higher-order terms $O(\|v\|^2)$.*

**Notation and preliminaries.** Throughout the proof, $\mathbf{r}_i \in \mathbb{R}^d$ denotes the projected residual vector in the parameter space spanned by $Q$. The subspace is equipped with the Fisher metric $\tilde{F} = \sum_{i=1}^{k} \tilde{F}_i$. For two $r$-dimensional subspaces $\mathcal{U}$ and $\mathcal{V}$, $d(\mathcal{U}, \mathcal{V}) = \|\sin\Theta(\mathcal{U}, \mathcal{V})\|_2$ denotes the principal angle distance.

The proof proceeds in three steps: (i) characterization of the minimum-disagreement subspace via the GEVP, (ii) proof of stability using matrix perturbation theory, and (iii) proof of identifiability via translation invariance.

### C.1. Step I: Characterization of the Minimum-Disagreement Subspace

We seek a consensus subspace $\mathcal{C}$ that minimizes the disagreement among experts relative to the local curvature. Recall from Eq. equation 16 that this is formulated as minimizing the Fisher-weighted residual energy projected onto the subspace:

$$\min_{B \in \mathbb{R}^{m \times r}} \operatorname{Tr}(B^\top \Sigma_r B) \quad \text{s.t.} \quad B^\top \tilde{F} B = I_r, \quad (16)$$

where $\Sigma_r = \frac{1}{k}\sum_{i=1}^{k} \mathbf{r}_i \mathbf{r}_i^\top$ is the empirical residual covariance, and the constraint $B^\top \tilde{F} B = I_r$ ensures the basis $B$ is orthonormal with respect to the Fisher metric $\tilde{F}$ (geometry-aware normalization).

The Lagrangian for this constrained optimization problem is:

$$\mathcal{L}(B, \Lambda) = \operatorname{Tr}(B^\top \Sigma_r B) - \operatorname{Tr}(\Lambda(B^\top \tilde{F} B - I_r)), \quad (17)$$

where $\Lambda$ is a symmetric matrix of Lagrange multipliers. Taking the derivative with respect to $B$ and setting it to zero

yields:

$$\frac{\partial \mathcal{L}}{\partial B} = 2\Sigma_r B - 2\tilde{F} B \Lambda = 0$$
$$\implies \Sigma_r B = \tilde{F} B \Lambda. \quad (18)$$

This is exactly the Generalized Eigenvalue Problem (GEVP) stated in Eq. equation 5. Since we wish to *minimize* the objective function $\operatorname{Tr}(B^\top \Sigma_r B) = \operatorname{Tr}(B^\top \tilde{F} B \Lambda) = \operatorname{Tr}(\Lambda) = \sum_{j=1}^{r} \lambda_j$, the optimal solution is given by the eigenvectors $\mathbf{b}_1, \ldots, \mathbf{b}_r$ corresponding to the $r$ *smallest* generalized eigenvalues $\lambda_1 \leq \cdots \leq \lambda_r$.

Thus, the consensus subspace $\mathcal{C}_r = \operatorname{span}\{\mathbf{b}_1, \ldots, \mathbf{b}_r\}$ characterizes the directions of minimum disagreement relative to the Fisher information.

### C.2. Step II: Uniqueness and Stability under Perturbations

Since $\tilde{F} \succ 0$, the GEVP is equivalent to the symmetric eigenvalue problem $A = \tilde{F}^{-1/2} \Sigma_r \tilde{F}^{-1/2}$, which preserves eigenvalues and invariant subspaces. Let $(\lambda_j, \mathbf{b}_j)$ be the eigenpairs of the pencil $(\Sigma_r, \tilde{F})$. Consider a perturbed covariance $\widehat{\Sigma}_r = \Sigma_r + \Delta$ with $\|\Delta\|_2 \leq \epsilon$. Assuming $\tilde{F}$ is positive definite (Assumption A.2), the pencil $(\Sigma_r, \tilde{F})$ is definite. This allows us to transform the GEVP into a standard symmetric eigenvalue problem for the matrix $A = \tilde{F}^{-1/2} \Sigma_r \tilde{F}^{-1/2}$.

Let $\mathcal{C}_r$ and $\widehat{\mathcal{C}}_r$ be the invariant subspaces corresponding to the $r$ smallest eigenvalues of the original and perturbed pencils, respectively. Applying the Davis-Kahan $\sin\Theta$ theorem (extended to definite matrix pencils (Pensky, 2024)), if there is a gap $\delta = \lambda_{r+1} - \lambda_r > 0$ separating the consensus spectrum from the unique value spectrum, the distance between the subspaces is bounded by:

$$d(\mathcal{C}_r, \widehat{\mathcal{C}}_r) \leq \frac{\|\tilde{F}^{-1/2}\Delta\tilde{F}^{-1/2}\|_2}{\delta} \leq \frac{\|\tilde{F}^{-1}\|_2 \|\Delta\|_2}{\lambda_{r+1} - \lambda_r}. \quad (19)$$

Recognizing that $\|\tilde{F}^{-1}\|_2 \approx 1/\lambda_{\min}(\tilde{F})$ relates to the condition number, we obtain the bound:

$$d(\mathcal{C}_r, \widehat{\mathcal{C}}_r) = O\left(\frac{\kappa(\tilde{F}) \cdot \epsilon}{\lambda_{r+1} - \lambda_r}\right).$$

This proves that the consensus subspace is stable to small estimation errors in the residuals, provided the distinct value directions (represented by larger $\lambda$) are sufficiently separated from the consensus manifold. The appearance of $\kappa(\tilde{F})$ reflects the conditioning of the metric induced by the Fisher geometry.

### C.3. Step III: Identifiability and Translation Invariance

We assume the Fisher information matrices $F_i(\theta)$ are locally Lipschitz and constant up to second order in a neighborhood

of $\theta_c^\star$. We verify that the consensus definition captures intrinsic geometry rather than artifacts of parameterization. Let the experts be shifted by a global vector $v$: $\theta_{\mathcal{A}}^{(i)\prime} = \theta_{\mathcal{A}}^{(i)} + v$.

Recall the PoE anchor is derived from $\nabla_\theta \mathcal{L}_{PoE} = 0$. The objective function for the shifted parameters is:

$$\mathcal{L}'(\theta) \approx \sum_{i=1}^{k} (\theta - (\theta_{\mathcal{A}}^{(i)} + v))^\top F_i (\theta - (\theta_{\mathcal{A}}^{(i)} + v)).$$

Let $\theta = \phi + v$. The gradient condition becomes identical to the original system, implying the new optimal anchor is simply shifted:

$$\theta_c^{\star\prime} = \theta_c^\star + v + O(\|v\|^2),$$

where the $O(\|v\|^2)$ term accounts for the higher-order variation in the Fisher matrix $F(\theta)$ (which is assumed constant locally under Assumption 3.2).

Consequently, the residuals remain invariant up to second order:

$$\begin{aligned} \mathbf{r}_i' &= \theta_{\mathcal{A}}^{(i)\prime} - \theta_c^{\star\prime} \\ &= (\theta_{\mathcal{A}}^{(i)} + v) - (\theta_c^\star + v + O(\|v\|^2)) \\ &= \mathbf{r}_i - O(\|v\|^2). \end{aligned}$$

Since the residuals $\mathbf{r}_i$ are invariant, the covariance $\Sigma_r$ and the resulting GEVP solution $\mathcal{C}_r$ remain invariant. Thus, the consensus subspace is an identifiable geometric object intrinsic to the set of aligned experts.

$\square$

# D. Extended Experimental Details

## D.1. Detailed Datasets Descriptions

We curate a multi-dimensional value spectrum by processing five datasets, each serving a specific role in validating our geometric alignment hypotheses.

**Dataset Description: The Moral Integrity Corpus (MIC)** We utilize the Moral Integrity Corpus (MIC), a large-scale resource introduced by Ziems et al. (2022), designed to benchmark the moral assumptions and social commonsense of open-domain dialogue systems. Unlike datasets focused on narrative text, MIC targets the minimal dialogue unit: a human *Prompt* and a system *Reply*.

**Data Collection and Source** The corpus consists of approximately 38,000 prompt-reply pairs. The prompts were sourced from the subreddit `r/AskReddit` to ensure a diverse range of open-ended, thought-provoking questions. To facilitate moral reasoning, the authors filtered for prompts

containing strongly subjective lexicon or terms related to moral foundations.

The system replies were generated using three distinct neural conversational models: BlenderBot, DialoGPT, and GPT-Neo. These pairs were subsequently filtered using a BERT-based classifier to ensure the replies were relevant, specific, and contained discernible moral or social content.

**Annotation Framework: Rules of Thumb (RoTs)** The core annotation of MIC is the "Rule of Thumb" (RoT), a free-text description of the moral principle that explains why a chatbot's reply is acceptable or problematic. The dataset contains approximately 99,000 distinct RoTs.

For each prompt-reply pair, crowd-workers provided:

- **Rule of Thumb (RoT):** A generative explanation of the underlying moral rule (e.g., *"You shouldn't judge people negatively based on their sexual orientation"*).

- **Alignment:** A categorical label indicating whether the chatbot's reply *agrees*, *disagrees*, or is *neutral* regarding the RoT.

- **Revised Answer:** A human-authored alternative reply that aligns with the generated RoT, enabling training for moral improvement.

**Moral Attributes and Foundations** To capture the nuance of moral judgment, MIC includes structured attributes for each RoT:

1. **Violation Severity:** A 5-point Likert scale ranging from *fine* (1) to *worst* (5), measuring the seriousness of breaking the rule.

2. **Global Consensus:** A measure of the perceived universality of the rule, ranging from *rarely* accepted to *universally* accepted.

3. **Moral Foundations:** Annotators classified RoTs into six moral foundations, extending the standard five-foundation theory to include Liberty:

   - *Care/Harm*
   - *Fairness/Cheating*
   - *Loyalty/Betrayal*
   - *Authority/Subversion*
   - *Sanctity/Degradation*
   - *Liberty/Oppression*

The final dataset comprises 114k structured annotations covering the 38k dialogue pairs. The train dev test split is organized such that no prompt-reply pair appears in multiple splits. The diversity of the dataset allows for the evaluation of models on both the generation of moral explanations and the classification of moral attributes.

**Daily Dilemmas: Navigating Value Conflicts** This dataset (Chiu et al., 2024) contains 1,360 vignettes featuring zero-sum moral trade-offs (e.g., Utilitarianism vs. Deontology).

- **Conflict Mapping:** We focus on primary conflict axes: *Care vs. Honesty* and *Fairness vs. Loyalty*.

- **Pareto Evaluation:** We utilize multi-dimensional labels to calculate scores across conflicting axes, allowing us to map the Pareto Frontier and assess how our geometric decomposition handles structural trade-offs compared to traditional interpolation.

**ValuePrism: Pluralism and Breadth** ValuePrism covers 218k values across 31k scenarios, making it the largest dataset for pluralistic alignment (Sorensen et al., 2024a).

- **Overton Window Evaluation:** We assess the model's ability to cover a broad "Overton Window" of socially acceptable opinions.

- **Analysis:** We test whether the learned geometric representations allow the model to output diverse perspectives on complex issues (e.g., "Privacy vs. Security") rather than collapsing into a single mode of "political correctness."

### D.2. Detailed Baseline Descriptions

To rigorously evaluate the efficacy of our proposed geometric consensus and steerable pluralism framework, we compare it against a diverse set of state-of-the-art baselines. These methods span three primary technical paradigms: model merging via task vectors, diversity-aware weight interpolation, and inference-time latent intervention. Below, we provide detailed formulations and implementation details for each baseline.

**Model Merging Methods** This category of baselines operates directly in the parameter space, aggregating multiple expert models—each fine-tuned for a specific value or preference—into a single aligned model.

**Fisher-Weighted Averaging (FWA) (Matena & Raffel, 2022).** FWA improves upon simple averaging by incorporating information about the local curvature of the loss landscape. It assumes a quadratic approximation of the loss, where the importance of each parameter is quantified by the diagonal of the Fisher Information Matrix (FIM), $F_i$. The merged parameters are computed as the precision-weighted average:

$$\theta_{\text{FWA}} = \left( \sum_{i=1}^{k} F_i \right)^{-1} \sum_{i=1}^{k} F_i \theta_{\mathcal{A}}^{(i)}. \tag{20}$$

By weighing parameters according to their "stiffness" (importance for the specific value), FWA mitigates catastrophic forgetting. However, unlike our approach, FWA does not perform explicit subspace decomposition to disentangle consensus from disagreement.

**DARE (Yu et al., 2024).** Drop And REscale (DARE) is a sparsification-based approach that exploits the redundancy in fine-tuned weights. DARE randomly drops (sets to zero) a large fraction $p$ of the delta parameters ($\tau_i$) and rescales the remaining parameters by a factor of $1/(1-p)$ to approximate the original function expectation.

$$\text{DARE}(\tau_i, p) = \frac{1}{1-p} \cdot m \odot \tau_i, \quad \text{where } m \sim \text{Bernoulli}(1-p). \tag{21}$$

Comparing against DARE validates the efficacy of our Fisher-weighted deterministic subspace selection against stochastic sparsification techniques.

**KnOTS (Stoica et al., 2025).** KnOTS (Knowledge Transfer via Singular Values) is an advanced merging technique that utilizes Singular Value Decomposition (SVD) to align the internal representations of models before merging. Recognizing that neurons in deep networks can be permuted or rotated without changing the function, KnOTS aligns the singular vectors of the expert models' weight matrices to a common basis to maximize overlap. This geometric alignment prior to averaging aims to reduce the "variance" in weight space caused by symmetries, offering a stronger baseline than direct weight averaging.

**Rewarded Soups (RS) (Rame et al., 2023).** Rewarded Soups leverages the phenomenon of Linear Mode Connectivity (LMC) to construct a "soup" of models that optimizes a specific reward function. Rather than a fixed average, RS performs a linear interpolation in weight space:

$$\theta_{\text{RS}} = \sum_{i=1}^{k} \alpha_i \theta_{\mathcal{A}}^{(i)}, \quad \text{s.t.} \sum \alpha_i = 1, \alpha_i \geq 0. \tag{22}$$

The coefficients $\alpha_i$ are optimized (usually via a gradient-free evolutionary strategy or lightweight gradient descent) to maximize a multi-objective reward signal on a validation set. This represents a strong "oracle" baseline for static alignment trade-offs.

**AlignMerge** (Roy et al., 2025) is a geometry-aware model merging framework designed to preserve the specific alignment properties (e.g., safety and helpfulness) of constituent models during fusion. Unlike standard linear merging techniques that often disrupt the delicate optimization landscape of aligned models, AlignMerge formulates the merging process as a constrained optimization problem

within the Fisher-Rao geometry. By estimating an alignment subspace around an instruction-tuned anchor and enforcing explicit geometric constraints, it ensures that the merged model retains critical alignment behaviors while effectively integrating diverse expert capabilities.

**MAP** (Wang et al., 2024b) formulates pluralistic value alignment as a constrained optimization problem, aiming to maximize a primary reward (e.g., helpfulness) while satisfying specific constraints on other objectives (e.g., safety or different value dimensions). To solve this, it employs a primal-dual approach, optimizing the Lagrangian function:

$$\min_{\boldsymbol{\lambda} \geq 0} \max_{\theta} \mathcal{L}(\theta, \boldsymbol{\lambda}) = \mathbb{E}_{\pi_\theta} \left[ R_0(x, y) + \sum_{k=1}^{K} \lambda_k (R_k(x, y) - \alpha_k) \right]$$
(23)

where $\theta$ represents the model parameters, $R_0$ is the primary reward, $R_k$ are the auxiliary rewards with corresponding safety thresholds $\alpha_k$, and $\boldsymbol{\lambda} = \{\lambda_1, \ldots, \lambda_K\}$ are the dual variables (Lagrange multipliers) that dynamically adjust the importance of each constraint based on its violation during training.

This category includes inference-time methods that steer the model without permanently merging weights.

**PAS (Personality Alignment Series) (Zhu et al., 2024).** PAS introduces a method for fine-grained steering based on personality traits. Unlike weight merging, PAS utilizes an efficient activation-level intervention. It leverages large-scale personality inventory data to learn "personality vectors" in the model's activation space. At inference time, these vectors are added to the hidden states of the LLM to shift its behavior toward a specific user preference profile (e.g., "Extroverted" or "Agreeable"). This serves as a primary baseline for our steering capability, representing activation-space steering versus our parameter-space steering.

### D.3. Detailed Experimental Setup and Evaluation Metrics

To strictly evaluate the steerability of the model across pluralistic values, we adopted a comprehensive evaluation framework consisting of three distinct target-setting regimes, a ground-truth determination protocol based on oracle attribute vectors, and rigorous training implementations.

**Alignment Target Regimes** Following prior work in Steerable Pluralism (Adams et al., 2025), we evaluated the model's ability to adapt to diverse and potentially conflicting value profiles using three distinct configurations of the target vector $\mathbf{v}$:

- **High Target Regime (Extreme Boundary Analysis):** In this setting, two specific alignment directions are

selected and explicitly optimized. The target vector is set to maximum intensity across two attributes, defined as $\mathbf{v}_{high} = [1.0, 1.0, 0.0 \ldots, 0.0]$. This regime assesses the model's ability to satisfy maximal constraints simultaneously and serves as a baseline for bias detection, ensuring the model does not default to high-value responses when low targets are requested.

- **Sampled Target Regime (Cardinality-Aware Stratification):** To evaluate fine-grained control, an independent target value is assigned to each value dimension, randomly sampled from the interval $(0, 1)$. To prevent performance on high-dimensional targets from overshadowing simpler tasks, we employed a stratified sampling strategy based on attribute cardinality. We uniformly sampled a fixed number of targets (e.g., 10) for each possible number of active attributes, ranging from sparse single attribute targets to dense full-attribute configurations.

- **All Target Regime (Comprehensive Evaluation):** This regime requires all value dimensions to be fully aligned, corresponding to an all-ones target weight vector.

**Accuracy Determination and Ground Truth** The "correct" response $k^*$ for a given input is not static; it is determined dynamically by comparing the user-specified target vector $\mathbf{v}$ against the oracle attribute vectors $\mathbf{t}_k$ associated with each candidate response $y_k$ $k = 3$ at ours datasets.

The ground truth is identified as the response that maximizes the similarity to the target vector:

$$k^* = \underset{k \in \{1, \ldots, K\}}{\operatorname{argmax}} \operatorname{sim}(\mathbf{v}, \mathbf{t}_k)$$
(24)

where $\operatorname{sim}$ denotes the similarity metric (e.g., Cosine Similarity or negative Euclidean distance). This protocol ensures that accuracy reflects the direction of the desired alignment profile rather than raw magnitude, making the metric robust to scale differences.

*Table 8.* Human Evaluation Statistics and Agreement Rates. This table reports the scale of the human evaluation and the inter-annotator agreement, verifying the reliability of the alignment measurement.

| Metric | MIC | Daily Dilemmas | Value Prism | Total / Avg. |
|---|---|---|---|---|
| *Annotation Scale* | | | | |
| Number of Annotators | 10 | 10 | 10 | 30 |
| Total Samples Evaluated | 50 | 50 | 50 | **150** |
| *Consistency Verification* | | | | |
| Inter-Annotator Agreement | 0.95 | 0.96 | 0.85 | **0.92** |
| Human-Model Align. Acc. | 0.91 | 0.88 | 0.83 | 0.873 |
| Label Consistency Rate | 0.92 | 0.83 | 0.81 | **0.85** |

**Human Evaluation and Consistency Verification** To complement automated metrics, we conducted human evaluation to verify the consistency between human judgments, model responses, and annotation labels. Table 8,give the details of human evaluation.

- **Objective:** The primary goal was to ensure that the generated model outputs aligned with the intended value dimensions as perceived by human evaluators, validating the reliability of the automated scale ratings.

- **Protocol:** The evaluation focused on the alignment accuracy by computing the similarity between model outputs and the corresponding value-dimension rating scales provided in the test set.

### D.4. Expert Training and Reward Modeling Implementation

We instantiated our framework using Llama-3.2-3B and Mistral-7B as backbone models. The training of value-specific experts followed well-established protocols using Direct Preference Optimization.

- **Training Objective:** Experts were trained using SFT, which implicitly optimizes the reward function by leveraging preference data $\mathcal{D}_i = \{(x, y_w > y_l)\}$ where $y_w$ is preferred over $y_l$ with respect to value $v_i$.

- **Hyperparameters and Optimization:**

  - **Optimizer:** The AdamW optimizer was used with a peak learning rate of $5 \times 10^{-5}$ and a cosine annealing schedule.
  - **Training Duration:** Each expert was trained for 3 epochs, with early stopping applied on a disjoint validation set to prevent overfitting.

- **Data Separation:** Training and evaluation data were strictly separated to ensure no leakage, guaranteeing that observed parameter differences arose solely from value-specific objectives.

- **Hardware:** All fine-tuning and manifold computations were performed on a cluster equipped with $3\times$ NVIDIA A800 (60GB) GPUs.

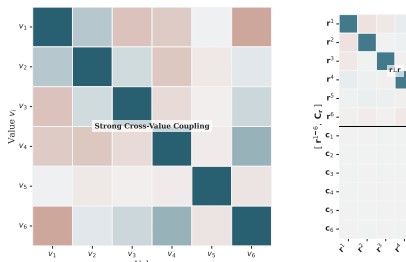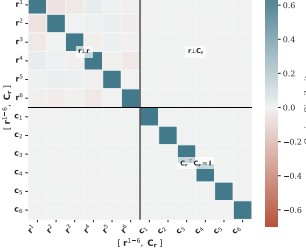

*Figure 9.* Value decoupling verification diagram before and after the method

## E. Add Experiments

### E.1. Diversified Alignment Experiment Supplement

We present the full and detailed experimental results in the table below, including all settings and evaluation metrics at Table 9.

In addition, we provide the detailed raw experimental data for the value-priority analysis below at Table 10.

### E.2. Low impact between values

We measure the geometric relationships between value representations on the MIC dataset before and after applying our method DisAlign . Prior to disentanglement, we observe that multiple value dimensions exhibit strong coupling, characterized by high pairwise similarities and significant directional overlap in the parameter space. This indicates that value experts trained on real-world data do not isolate individual values, but instead entangle multiple correlated value signals. As shown in Figure 9, After applying our disentanglement procedure, this coupling is substantially reduced. Specifically, the extracted value-specific subspaces become approximately orthogonal to the consensus subspace, demonstrating a clear separation between shared value components and value-unique representations. Moreover, the directions corresponding to different value-specific subspaces exhibit consistently low mutual correlations, indicating that each sub-value is encoded along an independent and non-interfering direction.

Taken together, these results confirm that our method effectively decomposes entangled value representations into a shared consensus space and multiple orthogonal value-specific subspaces. This structured decomposition enables independent modulation of individual values while preserving shared semantic foundations, thereby providing a principled geometric basis for precise and controllable pluralistic value alignment.

### E.3. More Ablation Study

**Minimal Degradation Alignment: Assessing the General Capability Tax**    A critical concern in model merging

and value steering is the potential degradation of the base model's general reasoning capabilities—often referred to as the *"Alignment Tax"* (Ouyang et al., 2022). While naive parameter intervention may successfully steer the model towards a specific value (e.g., "Helpfulness" or "Fairness"), it risks disrupting the delicate feature representations required for complex tasks such as mathematics and coding.

To rigorously evaluate whether DISALIGN incurs such a tax, we conducted a comprehensive evaluation on two distinct model architectures: **Llama-3.2-3B** and **Mistral-7B**. We assessed performance across four standard benchmarks covering diverse cognitive domains: MMLU (General Knowledge) (Hendrycks et al., 2020), ARC-CHALLENGE (Logical Reasoning) (Clark et al., 2018), GSM8K (Mathematical Reasoning) (Cobbe et al., 2021), and HUMANEVAL (Code Generation) (Chen, 2021).

**Results and Analysis.**    The quantitative results are presented in Table 11. We observe three key phenomena:

**1. The Cost of Naive Steering (Alignment Tax).** Consistent with the interference hypothesis, naively injecting value vectors causes catastrophic forgetting in specialized domains. As shown in Table 11, MISTRAL-7B suffers a massive degradation in mathematical ability, with GSM8K accuracy plummeting from 52.2% to 28.4% (−45.6% relative drop). Similarly, Llama-3.2-3B sees its reasoning capability (ARC-C) drop from 69.1% to 49.3%. This confirms that value-specific gradients, when not disentangled, destructively interfere with the model's reasoning circuits.

**2. DisAlign Enables Non-Destructive Steering.** In stark contrast, DISALIGN effectively circumvents this trade-off. By projecting value preferences onto a subspace orthogonal to the consensus manifold, our method preserves the base model's capabilities within a marginal variance (∼2.8% average drop). Notably, on MISTRAL-7B, DisAlign recovers the GSM8K score to 50.8%, demonstrating that it is possible to achieve strong value alignment (MIC High: 66.7%) without paying the alignment tax.

**3. Scalability Across Architectures.** The benefits of our geometric decomposition are consistent across model scales. Whether on the compact Llama-3.2-3B or the stronger Mistral-7B, DisAlign consistently outperforms naive baselines in both alignment controllability and capability preservation. This suggests that the *consensus-value* geometric structure is a universal property of aligned LLMs, robust to architectural differences.

In summary, the ablation study on general capabilities provides strong empirical evidence that the consensus subspace $\theta_c^*$ and $C$ successfully encode the model's fundamental reasoning skills, while the orthogonal value subspace $U_v$ isolates alignment preferences. This structural separation is the

*Table 9.* Evaluation of steerable pluralistic alignment on **MIC**, **Daily Dilemmas**, and **ValuePrism**. Metrics follow the Steerable Pluralism framework: **A** – Two-dimensional Accuracy, **B** – Random-dimensional Accuracy Accuracy, **C** – Full-dimensional Accuracy (↑). **Bold** indicates the best per metric. The second best results are indicated with underlining. Results are averaged over three runs with different seeds; standard deviations are $0.5 \sim 0.8$.

| Backbone | Method | MIC (Moral Foundations) | | | Daily Dilemmas (Preference) | | | ValuePrism (Values) | | |
|---|---|---|---|---|---|---|---|---|---|---|
| | | A. Two D. | B. Random D. | C. Full D. | A. Two D. | B. Random D. | C. Full D. | A. Two D. | B. Random D. | C. Full D. |
| | *Inference-time & Prompting-based Methods* | | | | | | | | | |
| Llama-3.2-3B | Steerable Pluralism | 53.2 | 52.4 | 53.5 | 51.5 | 53.6 | 51.8 | 52.4 | 51.2 | 52.1 |
| | Modular Pluralism | 55.8 | 53.2 | 52.5 | 54.3 | 53.8 | 52.2 | 52.8 | 52.8 | 53.3 |
| | Prompt-Align | 56.1 | 54.3 | 54.2 | 57.9 | 58.7 | 56.1 | 57.4 | 52.8 | 55.8 |
| | *Fine-tuning based Method* | | | | | | | | | |
| | MAP | 54.2 | 49.8 | 53.5 | 53.1 | 54.2 | 52.8 | 52.5 | 53.1 | 52.2 |
| | CPO | 53.7 | 50.3 | 52.8 | 53.8 | 51.9 | 52.6 | 52.9 | 50.1 | 49.7 |
| | *Model-merging based Method* | | | | | | | | | |
| | PAS | 53.5 | 48.5 | 53.1 | 56.2 | 58.5 | 54.9 | 55.1 | 56.8 | 54.2 |
| | Pmol | 50.4 | 50.3 | 50.5 | 49.9 | 53.8 | 51.5 | 51.9 | 52.8 | 51.1 |
| | Rewarded Soups | 50.8 | 42.5 | 51.2 | 49.1 | 41.2 | 50.5 | 48.9 | 40.8 | 50.1 |
| | FWA | 51.7 | 46.3 | 50.1 | 48.6 | 49.1 | 52.9 | 50.3 | 51.9 | 52.6 |
| | DARE | 50.5 | 51.8 | 51.0 | 50.5 | 49.6 | 50.6 | 51.2 | 52.3 | 50.3 |
| | AlignMerge | 51.3 | 51.2 | 49.3 | 48.9 | 47.1 | 51.6 | 50.1 | 48.9 | 48.7 |
| | KnOTS | 51.5 | 49.5 | 51.8 | 50.1 | 49.9 | 51.2 | 49.8 | 43.2 | 51.9 |
| | VISPA | 53.2 | 49.7 | 53.5 | 53.4 | 51.6 | 51.9 | 52.6 | 52.6 | 51.7 |
| | DisAlign(Ours) | **58.1** | **59.6*** | **56.6*** | **59.0*** | **60.8*** | **57.1** | **57.5** | **58.8*** | **56.2** |
| | *Inference-Time Steering & Prompt* | | | | | | | | | |
| Mistral-7B | Steerable Pluralism | 53.5 | 52.4 | 53.6 | 51.7 | 53.4 | 51.4 | 52.2 | 51.5 | 51.9 |
| | Modular Pluralism | 55.1 | 53.4 | 51.7 | 55.4 | 56.1 | 54.3 | 55.3 | 53.5 | 51.4 |
| | Prompt-Align | 56.7 | 63.9 | 55.9 | 56.3 | 58.6 | 55.4 | 55.9 | 59.8 | 55.5 |
| | *Latent Optimization & Neuron Guidance* | | | | | | | | | |
| | MAP | 54.8 | 50.5 | 53.9 | 53.5 | 54.8 | 52.2 | 53.1 | 51.2 | 52.5 |
| | CPO | 51.2 | 49.1 | 51.9 | 49.7 | 54.8 | 52.3 | 56.1 | 52.8 | 53.4 |
| | MODPO | 51.8 | 44.1 | 52.2 | 50.8 | 43.1 | 51.5 | 50.1 | 42.2 | 51.2 |
| | *Model Merging & Task Vectors* | | | | | | | | | |
| | PAS | 53.9 | 49.2 | 53.5 | 56.8 | 59.1 | 55.2 | 55.8 | 57.5 | 54.8 |
| | Pmol | 51.8 | 44.1 | 52.2 | 50.8 | 43.1 | 51.5 | 50.1 | 42.2 | 51.2 |
| | Rewarded Soups | 51.5 | 43.8 | 51.8 | 50.2 | 48.6 | 51.1 | 49.5 | 41.6 | 50.8 |
| | FWA | 48.4 | 47.5 | 51.9 | 52.4 | 42.9 | 45.8 | 50.3 | 49.6 | 45.2 |
| | DARE | 51.2 | 46.9 | 51.9 | 50.5 | 42.9 | 51.3 | 49.8 | 41.9 | 50.9 |
| | AlignMerge | 49.1 | 46.9 | 50.3 | 51.2 | 44.5 | 44.2 | 49.9 | 48.5 | 46.3 |
| | KnOTS | 52.2 | 44.5 | 52.5 | 51.2 | 47.5 | 51.9 | 50.5 | 42.6 | 51.6 |
| | VISPA | 54.2 | 51.7 | 53.9 | 54.8 | 52.7 | 53.9 | 54.6 | 53.6 | 52.7 |
| | DisAlign(Ours) | **58.5** | **66.7*** | **56.7** | **59.2*** | **64.2*** | **57.3*** | **58.2** | **65.5*** | **56.8** |

key to DisAlign 's ability to minimize the alignment tax.

## E.4. Hierarchy of Fisher Information Matrix Approximations

Let $\theta \in \mathbb{R}^d$ denote the model parameters and $\mathcal{D}$ be the alignment dataset. The exact empirical Fisher Information Matrix (FIM) is defined as the expected covariance of the score function:

$$F(\theta) = \mathbb{E}_{x,y \sim p_\theta} \left[ \nabla_\theta \log p_\theta(y|x) \nabla_\theta \log p_\theta(y|x)^\top \right]. \quad (25)$$

In practice, we estimate this using a Monte Carlo approximation over a finite dataset of size $N$, denoted as $\hat{F} = \frac{1}{N} \sum_{n=1}^{N} g_n g_n^\top$, where $g_n = \nabla_\theta \log p(y_n|x_n)$. Given the prohibitive dimensionality of Large Language Models

$(d \approx 7 \times 10^9$ for 7B models), storing or inverting the full matrix is computationally infeasible. To overcome this, we analyze the following hierarchy of approximation schemes used in our baselines and proposed method:

- **Diagonal Approximation (Diag-F):** This method assumes parameter independence, retaining only the diagonal elements: $F_{\text{diag}} = \text{diag}(\hat{F})$. This is the standard approach employed in Fisher-Weighted Averaging (FWA) (Matena & Raffel, 2022; Soen & Sun, 2024). While computationally efficient with $\mathcal{O}(d)$ complexity, Diag-F completely discards covariance information. Consequently, it fails to capture the correlations between parameters that are essential for characterizing the "entanglement" between different value dimensions

*Table 10.* **Joint Evaluation of Dual-Value Alignment under Priority Reweighting.** We measure alignment scores on both value dimensions $A$ and $B$ before and after priority changes. **Exp Full inversion experiment** evaluates coarse priority inversion from (Care,Loyalty) $(w_A, w_B) = (0.01, 1.0)$ to $(1.0, 0.01)$ MIC dataset. Unaliged model results is **40.3/39.1 Exp Fine-grained sensitivity experiment** evaluates fine-grained rebalancing in the ambivalence regime from $(0.5, 0.7)$ to $(0.7, 0.5)$. Effective controllability requires monotonic improvement on the promoted value while avoiding collapse on the demoted one. We given Non-Target Drift (NTD) and Fine-Grained Control (FGC) to assess whether they possess the desired alignment capabilities.

| Method | Exp A: Full inversion experiment | | | | | Exp B:Fine-grained sensitivity | | | | |
|---|---|---|---|---|---|---|---|---|---|---|
| | $A_{before}$ | $B_{before}$ | $A_{after}$ | $B_{after}$ | NTD | $A_{before}$ | $B_{before}$ | $A_{after}$ | $B_{after}$ | FGC |
| KnOTS | 46.8 | 56.4 | 55.6 | 44.8 | ✓ | 47.5 | 46.3 | 47.1 | 46.1 | × |
| Rewarded Soups | 47.2 | 55.8 | 56.1 | 43.3 | | 48.1 | 45.3 | 48.2 | 45.6 | × |
| MAP | 41.9 | 57.5 | 56.2 | 42.9 | ✓ | 49.2 | 50.1 | 50.8 | 49.9 | ✓ |
| **DisAlign** | 40.2 | 56.7 | 55.3 | 39.9 | No Drift | 49.7 | 50.4 | 50.9 | 48.1 | ✓ |

*Table 11.* **Cross-Model Evaluation of Alignment Tax and Efficacy.** We extend our analysis to LLAMA-3B to validate the scalability of DisAlign . **Crucially**, model possesses strong inherent mathematical and coding capabilities. Naive merging methods (represented by $\theta_0 + U_v\alpha$ ) incur a heavy "alignment tax," causing these capabilities to collapse. **DisAlign** consistently preserves these critical skills (within 2-3% variance) while achieving significantly higher controllability (MIC High Target) compared to baselines across both model architectures. Base model benchmarks are sourced from official technical reports.(llama3.2-3B)

| Backbone | Method / Configuration | General Capabilities (Acc. %) | | | | Alignment (MIC) | | Avg. Tax ↓ |
|---|---|---|---|---|---|---|---|---|
| | | MMLU | ARC-C | GSM8K | HUMANEVAL | Shared | Special parts | |
| **Llama-3.2 3B** | $\theta_0$ (Base Model) | **63.1** | **78.1** | 77.5 | 61.4 | 67.1[†] | 30.7 | - |
| | $\theta_c^* + \beta C_r$ (Consensus Only) | 62.8 | 76.5 | 75.9 | 60.1 | 94.8 | 35.4 | 1.7% |
| | $\theta_0 + U_v\alpha$ | 46.5 | 50.3 | 60.2 | 45.5 | 62.1 | 54.7 | 27.5% |
| | **DisAlign (Ours)** | 62.1 | 77.2 | **77.6** | **61.8** | **95.6** | **53.1** | **0.5%** |

[†] Random/Unsteered baseline performance. CODE refers to HumanEval pass@1. Llama-3.2 ARC-C is higher than Mistral due to architectural optimization for reasoning. Alignment scores for Naive methods are averaged from FWA/DARE results in Table 6.

(e.g., how modifying *Honesty* impacts *Helpfulness*), making it unsuitable for the precise disentanglement required in DisAlign .

- **Block-Diagonal Approximation (Block-F):** This scheme assumes independence between layers but retains full dense covariance within each layer (or attention head) (Soen & Sun, 2024). Let the parameter vector be partitioned into $L$ blocks $\{\theta_l\}_{l=1}^L$. The matrix takes the form $F_{\text{block}} = \text{blkdiag}(F_1, \ldots, F_L)$, where $F_l \in \mathbb{R}^{d_l \times d_l}$ corresponds to the dense Fisher block of the $l$-th layer. In our experiments, we utilize this as an "Oracle" baseline for local geometry. Although it captures intra-layer correlations, estimating and inverting high-dimensional layer blocks remains computationally expensive and memory-intensive for large-scale models.

- **Subspace-Projected Fisher:** Instead of approximating the high-dimensional matrix directly (Chen & Kortje, 2025), computes the exact Riemannian metric restricted to the active consensus subspace $\mathcal{S}$. Let $Q \in \mathbb{R}^{d \times m}$ be an orthonormal basis matrix spanning the subspace of expert deviations (i.e., span$\{\theta_{\mathcal{A}}^{(i)} - \theta_0\}_{i=1}^k$).

The projected Fisher matrix $\tilde{F} \in \mathbb{R}^{m \times m}$ is computed as:

$$\tilde{F} = Q^\top \hat{F} Q = \frac{1}{N} \sum_{n=1}^N (Q^\top g_n)(Q^\top g_n)^\top. \quad (26)$$

Critically, this formulation reduces the computational problem from the ambient parameter space $\mathbb{R}^{d \times d}$ to the low-rank manifold $\mathbb{R}^{m \times m}$ ($m \ll d$). By computing the vector-Jacobian products $v_n = Q^\top g_n$ first, we avoid ever materializing the full $d \times d$ matrix. This allows us to capture the *exact* geometry and curvature interactions along the relevant directions of value variation, enabling the precise spectral decomposition described in Section 3.4.

### E.5. Sampling Strategy and Numerical Stability

**Sample Efficiency.** Consistent with prior work on empirical Fisher estimation (Matena & Raffel, 2022), we utilize a subset of samples $N$ from the alignment dataset to estimate the geometry. As demonstrated in Figure 10, the spectral properties of the estimated geometry converge rapidly. Because we operate in a low-dimensional projected space

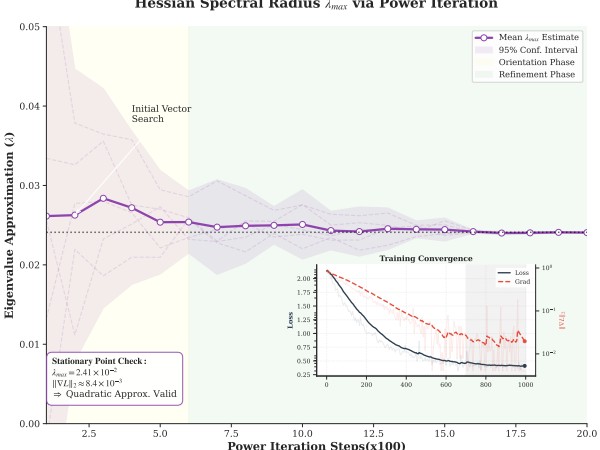

*Figure 10.* **Convergence of the Hessian Spectral Approximation.** We verify the stability of our Subspace-Projected Fisher estimation by monitoring the spectral radius via power iteration. The approximation error decreases rapidly as the number of samples $N$ increases, confirming that the low-rank subspace geometry can be efficiently characterized with a limited data budget.

defined by the $k$ experts ($m \leq k \ll N$), the sample covariance matrix is significantly better conditioned than high-dimensional estimations. This avoids the "curse of dimensionality" that typically plagues Fisher estimation in LLMs, allowing us to obtain a robust metric $\tilde{F}$ with as few as $N \approx 1024$ samples.

**Damping and Regularization.** To ensure numerical stability during the Generalized Eigenvalue Problem (GEVP) and matrix inversion steps, we apply standard Tikhonov regularization (damping) to the projected Fisher matrix:

$$\tilde{F}_{\text{reg}} = \tilde{F} + \gamma I_m, \tag{27}$$

where $\gamma = 10^{-5}$ is a small damping coefficient. This regularization prevents singularity in subspace directions where the empirical curvature might be near-zero due to limited sampling or parameter insensitivity (flat directions in the loss landscape), ensuring that the projection operator $\Pi_{\mathcal{C}_r}^F$ remains numerically stable.

### E.6. Stability Analysis of the Spectral Gap

In this section, we provide extended empirical analysis regarding the stability of the spectral gap and its impact on the estimated subspace dimension ($r_{\text{eff}}$) across various models and datasets. The results are summarized in Table 12.

**(1) A clear spectral gap appears in most cases.** Across the 9 dataset–backbone combinations evaluated, 7 combinations exhibit a strong spectral gap ($> 4$). Even for the remaining cases, the gap remains around 3. These results consistently support the stability and identifiability conditions outlined in Proposition 3.4.

**(2) A small spectral gap minimally affects our method's performance.** When the spectral gap is relatively small (e.g., on the ValuePrism dataset), the entropy criterion may slightly overestimate the effective subspace dimension (resulting in $r_{\text{eff}} > r$). However, because the leading components already capture the dominant value directions, the practical impact of this overestimation is negligible. Empirically, setting $r_{\text{eff}} = r$ on ValuePrism yields accuracy highly comparable to using the overestimated $r_{\text{eff}}$. For instance, with the Mistral-7B backbone, forcing $r_{\text{eff}} = r$ achieves an accuracy of 65.1, closely matching the 65.5 accuracy obtained with $r_{\text{eff}} = r + 2$. Both of these results significantly outperform the DARE baseline (41.9).

*Table 12.* Spectral gap and performance stability across different dataset-backbone combinations. A checkmark (✓) indicates $r_{\text{eff}} = r$. For cases where $r_{\text{eff}} > r$, the performance when forcing $r_{\text{eff}} = r$ is presented in parentheses.

| Dataset | Backbone | Spectral Gap | $r_{\text{eff}} = r$ | Ours | DARE |
|---------|----------|--------------|----------------------|------|------|
| MIC | Llama-3B | 10.2 | ✓ | 59.6 | 51.8 |
| MIC | Mistral-7B | 9.8 | ✓ | 66.7 | 46.9 |
| MIC | Qwen3.5-4B | 11.7 | ✓ | 62.4 | 50.1 |
| Dilemmas | Llama-3B | 9.6 | ✓ | 60.8 | 49.6 |
| Dilemmas | Mistral-7B | 8.5 | ✓ | 64.2 | 42.9 |
| Dilemmas | Qwen3.5-4B | 10.8 | ✓ | 62.5 | 44.3 |
| ValuePrism | Llama-3B | 3.4 | ✓ | 58.8 | 52.3 |
| ValuePrism | Mistral-7B | 4.1 | $r_{\text{eff}} = r + 2$ | 65.5 (65.1) | 41.9 |
| ValuePrism | Qwen3.5-4B | 2.8 | $r_{\text{eff}} = r + 3$ | 63.8 (62.3) | 47.8 |

