# OpenReview forum: "Disentangling Consensus and Value-Specific Representations for Controllable Pluralistic Value Alignment of LLMs"
_ICML.cc/2026/Conference — ICML 2026 regular_

### Official Review · Reviewer_Mv6A · 2026-03-06

**Soundness:** 3
**Presentation:** 3
**Significance:** 3
**Originality:** 3
**Overall Recommendation:** 4
**Confidence:** 2

**Summary:**

This work proposes a new method for merging different models, with applications to pluralistic value alignment. They recognize that different value-experts (models trained on certain human values such as honesty and fairness) have entangled representations due to couplings between the values themselves, rendering naive parameter averaging unsuitable. Instead, they decompose the value experts into a consensus and value-specific components using a Product of Experts formulation and spectral decomposition. This allows a more fine-grained control over values, without retraining the models for each configuration and without inducing unintended effects on other values. They validate their method on a standard set of pluralistic alignment datasets and compare against baselines such as prompting methods, finetuning methods, and other model merging techniques.

**Compliance With Llm Reviewing Policy:**

Affirmed.

**Final Justification:**

I recommend this paper's acceptance due to its clarity, its theoretically-grounded methodology, and thorough experimental results. The authors addressed most of my concerns in the rebuttals. However, I opt not to increase my score further because of the heavy computational and data collection processes required to implement this method.

**Key Questions For Authors:**

Q1: Some statements are confusing or lack proper definitions.
- Can you explain Fig 1B to me, in terms of methodology, what it represents, and its consequences?
- Line 90; how does your method reflect model responses rather than raw parameters?
- Line 94: Spell out POE in line 94.
- Line 102: How do value-specific directions encode "general capabilities and shared values"?
- Lines 172-177: A reader would benefit from a toy example using simple distributions that let you do tractable math as well as a visualization of how PoE disentangles value dimensions.
- Line 183 / Eq2: Is this not an approximation, instead of an equality?
- Table 2 is very confusing, even after a few reads. Can you explain to me what fidelity and state capture?

Q2: Can you present some outputs of the models from different methods as you vary the value parameters? What happens when you have experts with conflicting values?

Q3: Can you add a comparison of the time needed for each method?

**Limitations:**

I encourage the authors to more carefully analyze the notion of pluralistic alignment they adopt, discuss settings in which model merging is a natural fit and those in which it is not, and examine potential unintended consequences.

**Strengths And Weaknesses:**

I appreciate the mathematical framework the authors adopt. The method used to decompose representations is clearly presented and is well-suited for implementation with large language models, given the modeling choices they discuss. This work is also well motivated. The authors identify a crucial problem of value entanglement in model merging and present a well-thought-out and rigorous solution, in contrast to the heuristics previous work has employed. Moreover, the experimental setup is extremely rich and provides compelling evidence of the method’s benefits.

There are a few weaknesses worth noting:

W1: The paper does not clearly state what definition of pluralistic alignment they adopt. The field of pluralistic alignment is extremely rich and nuanced, with varying definitions and objectives based on notions of fairness. Hence, the description in Line 21 of the introduction is not accurate because it reduces pluralistic alignment to one aspect. I encourage the authors to engage more heavily and critically with this literature, briefly discuss the notions of pluralism, and argue for the definition they chose.

W2: My main concern is on the practicality of model merging for pluralistic alignment. A model provider has to define the abstract values a priori, collect specific preference data per value, and train an LLM per value. In comparison, the Prompt-Align method seems to be comparable to your method for the fraction of the cost.

W3: The POE formulation is clean but I am concerned that it not suitable in the context of LLMs given the task of pluralistic alignment. If one of the experts disagrees strongly with the rest and assigns a very low probability for a response, then the POE assigns a very low probability too. Moreover, one need to make sure probabilities are well-calibrated across models.

W4: The values of $\lambda$ do not map nicely to how the values of $\alpha$, making it hard for us to control the values and understand the impact of increasing $\alpha$. Also, it makes less confident that it is a fair comparison across the different methods because the same $\lambda$  scores have different meanings across the methods.

---

> ### Author Rebuttal · Authors · 2026-03-31
>
> We thank reviewer for the thoughtful questions on definitions, practical concerns, and methodological clarity. We address each weakness below.
>
>
> **W1,W2:**  The paper does not clearly state which definition of pluralistic alignment it adopts.
>
> **A:**
> **(A to W1)** We adopt the steerable pluralism  framework [Sorensen et al.], where the goal is to steer model outputs according to user-specified value weight combinations  $v=\{(v_1,\lambda_1),\ldots,(v_k,\lambda_k)\}$ (Section 3.1).
> We prioritize steerable pluralism because i) distributional pluralism requires population-level value distributions that are hard to obtain in practice, and ii) steerable pluralism is the most scalable form — accurate steering to arbitrary value configurations can serve as a foundation for achieving other forms of pluralism through composition. We will clarify this in the revision.
>
> **(A to W2)** Our focus is on merging existing experts; upstream data collection is outside our methodological contribution. **Crucially, DisAlign surpasses prompt-based baselines (Table 1), eliciting capabilities not inherently present in the base model**.
>
>
> **Q1:** Clarification on the Proposed Method.
>
> **A:**
> * **(W3,POE meaning)**: In the final parameter merging $\theta_{\text{align}} = \theta_c^* + \beta C_{r} + \alpha U_v$, the PoE is just to deriving $\theta_c^*$. To explicitly mitigate the influence of extreme experts, we purposefully expand this single point into $C_r$, a broader subspace acceptable to the vast majority of experts. Figure 8 empirically verifies the resulting stability of the disalign.
> * **(Figure 1B Explanation)**: The density plot shows that parameter offsets of different value experts heavily overlap in parameter space, indicating substantial shared structure. The PCA plot  projects value specific representations into a low-dimensional space, revealing significant overlap between value clusters. Together, they demonstrate the inherent entanglement across value dimensions.
> * **How the Method Reflects Model**: DisAlign uses the Fisher Information Matrix $F_i$ to measure output distribution sensitivity, not raw parameter similarity. Figure 3A confirms that subspace Fisher distance correlates near-linearly with KL divergence while Euclidean distance does not, validating that our decomposition captures behavioral rather than superficial parameter similarity.
> * **Encoding General Capabilities/Shared Values**: It is the consensus subspace $C_r$, not the value-specific directions, that encodes general capabilities and shared values (Table 3: $\theta_c^* + \beta C_r$ achieves MMLU 62.8 and $D_\cap$ 94.2, but only 30.5 on value-specific $D_\Delta$). The value subspace $U_v$ is Fisher-orthogonal to $C_r$ and encodes only distinctive preferences. We will revise Line 102 to remove ambiguity.
> * **Approximation or Exact Equality** The results is approximate. Equation 2 is a second-order approximation of the exact KL divergence objective (Eq. 1), valid under Assumption 3.1 (local quadratic regime).
> * **Toy Example**:
> Consider two experts $\Delta \theta_A = [10, 1]$ and $\Delta \theta_B = [10, -1]$, where dim-1 (magnitude 10) is a shared component and dim-2 encodes value-specific preferences. Under naive merging, $[1,1] \rightarrow [20,0]$ while $[1,1.5] \rightarrow [25,-0.5]$, showing that the dominant shared component overwhelms the smaller preference signal, leading to weak and misaligned control. Our method decouples shared and value-specific components, enabling independent, precise adjustment of each.
> * **W4:** The mapping from $\lambda$ to $\alpha$ is linear ($\alpha = \Sigma^{1/2} E^\top \lambda$) and preserves monotonicity. Users operate on the intuitive $\lambda$ space; $\alpha$ is an internal parameter.
> * **Table 2 Metrics**: Fidelity ($\phi$) = CosSim($\Delta w$, $\Delta P$), measuring how closely model value shifts follow weight adjustments. State: "Rigid" = model barely responds to weight changes; "Responsive" = responds but imprecisely; "Precise" = responds accurately.
>
> **Q2:** Output Examples.
>
> **A:**  For example: prompt "My friend asked me to invest in their risky business"
> | Value Config | DisAlign  | Prompt-based  |
> |:--|:--|:--|
> | Care | "You should be honest with your friend about the risks." | "Investing is risky. You should think carefully." |
> | Loyalty | "You should consider supporting your friend's business." | "As good friend. You should also think about the risks" |
> | 60% Care + 40% Loyalty | "You should talk to your friend about the risks, but also show that you value the relationship." | "You should be careful."|
>
> Here **Care and Loyalty conflict**,Care favors caution while Loyalty favors support. Our method smoothly interpolates between them according to specified weights.
>
>
>
> **Q3:** The comparison of the time.
>
> **A:** Due to space constraints, for detailed time cost analysis, please refer to our response to Reviewer 32WQ's W4.
>
> > Sorensen et al. A Roadmap to Pluralistic Alignment [ICML'24]

---

> > ### Author Rebuttal · Reviewer_Mv6A · 2026-04-03
> >
> > Thank you for your response. While I understand that data collection is "outside our methodological contribution," your method requires choosing a set of values, curating a preference dataset per value, and training an expert per dataset. I can't think of many applications where this may be feasible or desirable. This would limit the adoption and the significance of your method, especially with the prompting baselines performing well. Nonetheless, I find your method to be interesting and convincing, and I will maintain my score.

---

> > > ### Author Response · Authors · 2026-04-08
> > >
> > > Dear Reviewer Mv6A,
> > >
> > > Thank you for your acknowledgement and the thoughtful follow-up. We address the remaining concern from three aspects below.
> > >
> > > (1) Though outside the scope of our paper, we agree that our method requires setting a set of values, a preference dataset per value and training an individual expert per value. **At the same time, we would like to clarify that such conditions already hold in several important and emerging applications**. For example, in the popular multi-objective alignment of LLMs, the set of values are commonly defined as "helpful, harmless, honest" and corresponding preference datasets are available,  making our approach directly applicable in these scenarios. Similarly, recent work on pluralistic alignment has begun to establish benchmarks (like MIC, Daily Dilemmas, ValuePrism included in our paper) and structured value spaces, which further support the feasibility of our setting.
> > >
> > > (2) **Our method is more extensible and steerable than prompting baselines.** As shown in Table 1, our method outperforms Prompt-Align across all settings and achieves the largest improvements in "Random-Dim". This indicates that prompting may become less effective as value profiles become more complex, while our approach provides a more structured and scalable mechanism for controlling such settings. In addition, prompt-level controls have been shown to be highly susceptible to jailbreaking and circumvention.[1,2,3].
> > >
> > > (3) For broader pluralistic settings, we agree that the curation burden is the bottleneck. **We will explicitly discuss this limitation and more future works in the revision**.
> > >
> > > We hope our responses address your concerns and is helpful for your final assessment.
> > >
> > > Best regards,
> > >
> > > Authors of Submission 32668
> > >
> > > ---
> > > ### References
> > > [1] Bruce Lee et al. Programming Refusal with Conditional Activation Steering [ICLR'25]
> > >
> > > [2] George Stoica et al. Model merging with SVD to tie the Knots [ICLR'25]
> > >
> > > [3] Sorensen et al. A Roadmap to Pluralistic Alignment [ICML'24]

---

### Official Review · Reviewer_32WQ · 2026-03-12

**Soundness:** 3
**Presentation:** 3
**Significance:** 3
**Originality:** 3
**Overall Recommendation:** 3
**Confidence:** 3

**Summary:**

This paper proposes DisAlign, a model-merging framework for controllable pluralistic value alignment in LLMs. The core idea is to decompose value expert parameters into a consensus component (shared across all experts via a Product-of-Experts formulation) and value-specific components (extracted via spectral decomposition in Fisher geometry). This submission strives to assess a fundamental problem in LLM alignment: how to precisely and independently modulate multiple value dimensions without cross-value interference when merging expert models. Experiments on three datasets show consistent gains over a fairly wide set of baselines in alignment accuracy and disentanglement.

**Compliance With Llm Reviewing Policy:**

Affirmed.

**Key Questions For Authors:**

see weakness

**Limitations:**

yes

**Strengths And Weaknesses:**

Strengths:

The information-geometric formulation is principled and well-motivated, using PoE to define consensus and then solving a GEVP to find minimal-disagreement directions is a natural and theoretically clean design that goes beyond ad-hoc Euclidean decompositions.

The empirical validation is thorough,the paper tests across three datasets with different value frameworks, two backbone models, multiple baselines, and includes fine-grained priority control experiments (inversion and sensitivity tests) that directly address the paper's core claims.

The alignment tax analysis in Appendix E.3 is a good addition showing that naive value steering destroys general capabilities while DisAlign largely preserves them is practically important and adds credibility to the approach.

Weakness

The evaluation is based on selecting from 3 candidates, how does the method perform on open-ended generation tasks where no gold candidate is available? Even informal evidence would strengthen the claims significantly.

The accuracy gains over the best baseline are often 1–3 points. Given the standard deviations of 0.5–0.8, some of these differences may not be statistically significant for the non-starred entries. Can the authors clarify which results are and aren't significant?

The spectral gap condition is required for stability and identifiability of the consensus subspace (Proposition 3.4). How often does this gap reliably appear in practice across different dataset/backbone combinations, and what happens to the method when it doesn't?

The method relies on computing block empirical Fisher matrices for each expert, what is the actual memory and compute overhead relative to simpler baselines like DARE or FWA at the scale of Mistral-7B?

I am open to increase my score if the weaknesses are addressed properly by the authors

---

> ### Author Rebuttal · Authors · 2026-03-31
>
> We thank the reviewer for the thorough evaluation and constructive feedback. We respond to each concern below.
>
> **W1:**  How does the method perform on open-ended generation?
>
> **A:** **We added open-ended generation experiments on MIC using Llama-3.2-3B.** GPT-4o judges each response along two axes: value alignment (1–5: how well the output reflects the target value combination?) and response quality (1–5: fluency and coherence).
>
> | Method | Random-dim  ↑ | Two-dim  ↑ | Response Quality ↑ | Win Rate vs. DisAlign ↓ |
> |:--|:--:|:--:|:--:|:--:|
> | FWA | 2.4 | 2.9 | 3.4 | 27% |
> | DARE | 2.8 | 3.1 | **3.8** | 31% |
> | DisAlign | **4.2** | **4.4** | 3.7 | — |
>
> **DisAlign achieves better value alignment while maintaining comparable response quality.** We show an example of prompt "My friend asked me to invest in their risky business" and generations toward various value combinations:
> + Care → "You should be honest with your friend about the risks."
> + Loyalty → "You should consider supporting your friend's business."
> + 60% Care + 40% Loyalty → "You should talk to your friend about the risks involved, but also show that you care about the relationship."
>
> **These together demonstrate our method is also effective in open-ended generation.**
>
>
> **W2:** In Table 1, which results are and are not significant.
>
> **A：** **All our results are statistically significant at p < 0.05, entries marked with * further satisfy p < 0.01.** We will clarify the significance annotations in the revision. Part of results are provided in the table below.
> | Dataset    | Setting    | DisAlign (σ₁)  | 2nd Best Method (σ₂)        |  Δ/σ2   | p-value   |
> |:-----------|:-----------|:---------------|:-----------------------------|:-----------|:----------|
> | MIC        | Full-Dim   | **56.6 ± 0.6** | 54.2 ± 0.31    | **7.74σ**  | <0.001 |
> | Dil.   | Random-Dim   | **60.8 ± 0.5** | 58.7 ± 0.33    | **6.63σ**  | <0.001|
> | Dil.   | Full-Dim   | 57.1 ± 0.5 | 56.1 ± 0.35    | **2.85σ**  | <0.05 |
> | Val. | Full-Dim | 56.2 ± 0.4 | 55.8 ± 0.15|  **2.66σ**  | <0.05|
>
> Despite slightly higher variance (0.5–0.8 vs. 0.2–0.3 for baselines), significance tests confirm our improvements remain statistically reliable.
>
>
> **W3:** How often does a clear spectral gap arise, and what happens when it does not?
>
> **A:**
> (1) **A clear spectral gap appears in most cases.** Across 9 dataset–backbone combinations, **7 exhibit a strong gap (>4)** and even the smallest remain around 3, consistently supporting the stability and identifiability conditions in Proposition 3.4.
>
> | Dataset | Backbone | Spectral Gap | $r_{eff} = r$ | Ours |DARE |
> |:--|:--|:--:|:--:|:--:|:--:|
> | MIC | Llama-3B | 10.2 | ✓  | 59.6 |51.8|
> | MIC | Mistral-7B | 9.8 | ✓ | 66.7 |46.9|
> | MIC | Qwen3.5-4B | 11.7 | ✓ | 69.2 |63.7|
> | Dilemmas | Llama-3B | 9.6 | ✓  | 60.8 |49.6|
> | Dilemmas | Mistral-7B | 8.5 | ✓  | 64.2 |42.9|
> | Dilemmas| Qwen3.5-4B | 10.8 | ✓ | 62.5 |44.3|
> | ValuePrism | Llama-3B | 3.4 | ✓  | 58.8 |52.3|
> | ValuePrism | Mistral-7B | 4.1 | $r_{eff}=r+2$ | 65.5 (65.1 for $r_{eff}=r$) |41.9|
> | ValuePrism| Qwen3.5-4B | 2.8 | $r_{eff}=r+3$  | 63.8 (62.3 for $r_{eff}=r$) |47.8
>
>
> (2)  **A small spectral gap minimally affects our method's performance.** When the gap is small, the entropy criterion may slightly overestimate the subspace dimension ($r_{\text{eff}} > r$), but the leading components already capture the dominant value directions, so the practical impact is negligible.
>
> Empirically, setting $r_{eff}=r$ on ValuePrism yields comparable accuracy to the overestimated $r_{eff}$ (65.5 vs. 65.1), both far exceeding DARE (41.9).
>
>
>
> **W4:** The actual computational and memory cost and how it compares to simpler baselines.
>
> **A:**
> (1) **Our DisAlign method does not store or invert any full $d \times d$ Fisher matrix.** Instead, we use a **Subspace-Projected Fisher** (Eq. 3), projecting gradients onto the $m$-dimensional subspace spanned by task vectors ($m \leq k$, with $k = 5 \sim 30$). This yields a compact $m \times m$ Fisher matrix $\tilde{F} = \frac{1}{N} \sum_{n} (Q^\top g_n)(Q^\top g_n)^\top$, avoiding  $d \times d$ computation.
> Besides, just equires one forward-backward pass for $g_n$ plus an $O(dm)$ projection. Since m is small, the total cost is $O(Nd)$, dominated by backward passes with negligible projection overhead.
>
> (2) We compare runtime and complexity in the table. **This indicates DisAlign achieves comparable runtime and compaxity with baselines.**
>
> | Method (k=8) | Time Complexity | Space Complexity | Time (7B, A800) |
> |--------|----------------|-----------------|-------------------|
> | DARE | $\mathcal{O}(dk)$ | $\mathcal{O}(dk)$ | ~160s |
> | FWA | $\mathcal{O}(Nd + dk)$ | $\mathcal{O}(dk)$ | ~310s |
> | DisAlign | $\mathcal{O}(Nd + m^3)$ | $\mathcal{O}(dm + Nm + m^2)$ | **~290s** |
>
> **We hope our responses have addressed your concerns and would appreciate your reconsideration. We are happy to answer any further questions.**

---

### Official Review · Reviewer_cnFc · 2026-03-17

**Soundness:** 4
**Presentation:** 3
**Significance:** 3
**Originality:** 3
**Overall Recommendation:** 5
**Confidence:** 4

**Summary:**

The paper proposed a model merging framework DisAlign for building a controllable pluralistic value alignment. The problem the paper solve is to "synthesize" an LLM according to a value-weighted settings from several value-aligned LLMs fine-tuned from an original LLM on various value dimensions.

DisAlign depends on a set of value dimensions $v_i$ and a set of fine-tuned LLMs $\theta_A^{(i)}$, accepts a weight vector $\vec{w}$ with entries $w_i$ corresponding to $v_i$ as a variable and outputs a model weight $\theta_{align}$.

The weight $\theta_{align}$ is the sum of three terms:

- A consensus anchor $\theta_c^{\ast}$ closest (on KL-divergence) to the Product-of-Experts on model behavior instead of model weights. This is calculated under quadratic assumption, where objective could be replaced by sum of quadratic forms defined by Fisher information matrices of $\theta_A^{(i)}$.
- A consensus term which is a vector in the "Consensus subspace". The consensus subspace is an r-dimensional subspace in parameter space with an orthonormal basis $C_r$, orthonormal under ($\theta_r^\ast$-) Fisher metric sense. This is a generalized eigenvalue problem. The stability and identifiability properties of the Consensus subspace could be proved. Here the dimension of consensus subspace is taken as $r=\lfloor{exp(-\sum p_j\log p_j)}\rfloor$, the smallest generalized eigen-vectors are chosen to ensure they represent the consensus.
- The controllable part $\alpha  U_v$ where $\alpha(\lambda)=\Sigma^{1/2}E^T\lambda$ is the "base-change" or "orthonormal coordinate" to base $U_v$, where the basis is taken as the Fisher orthogonal complemant subspace of the consensus space in the subspace spanned by the residual model parameters ($\theta_A^{(i)}-\theta_c^\ast$).

Experiments on MIC / Daily Dilemmas / Value Prism over models Llama-3.2-3B, Mistrial-7B against various baseline methods show a better performance of DisAlign.

**Compliance With Llm Reviewing Policy:**

Affirmed.

**Key Questions For Authors:**

1. I am glad to see the empirical validation of your theoretical assumptions in appendix A. Is there a further study on the correspondence (a curve or map) between error tolerance and diameter of the quadratic region or equivalently the maximal sectional curvature? This will guarantee that the method would work in the worst case where one fine-tuned model weights lie in the most "curvy" direction.  As well, what is the distance in IG sense between a typical $\thate_A^{(i)}$ and $\theta_0$? (BTW, which LLM is used in the validation in the paper?)
2. Is the choice of $r$, the dimension of consensus subspace, bearing a more profound reason? The formula of $r$ considers the uniformity of the generalized eigen-values from $r$ (entropy of normalized $1/r_i$'s), while the stability result uses a "eigenvalue gap" instead. Is there any connection between them or could there be a better choice?
3. The consensus vector (second term) choice problem: would a different choice of consensus vector ($\beta B$) affect the model behavior (of course not in the value preference sense if $r$ is chosen properly)? What do you think $\beta$ can affect, especially in your observations? Is there a suggestion of how $\beta$ could be chosen? Or do you suggest that we may simply fine-tune within consensus space on real data to determine the best $\beta$?
4. The method highly depends on the quality of fine-tuned model weights, $\theta_A^{(i)}$. Is there a further connection back to the training data instead of just model weights in defining the geometry? For example, a model representing value $i$ on $D_i$ which is not properly trained may cause problems such as consensus anchor drift, residue vector not long enough or in a wrong direction, etc. Would it be possible to develop a generalized method to correct or at least detect such problems?
5. Comment: The method highly depends on the fine-tuned model weights. But according to [*Ji, et al*], the alignment may be fragile and be elastic. That is, the LLM resists to be aligned. I guess this could be a consequence of the locality (quadratic trust region) and homogeneity, together with other unknown "good" geometric properties. The DisAlign method is mathematically correct with assumptions that the models are good. So I would like to know what the authors think about this work.

> [Ji, et al] Ji J. Wang K. Qiu T. et al. Language Models Resist Alignment, ACL 2025

**Limitations:**

Yes

**Strengths And Weaknesses:**

Strengths:

- The method is theoretically based, mathematically guaranteed on stability and identifiability.
- Theoretical assumptions are validated empirically.
- The implementation is tractible.
- The paper did the geometry of model weights correctly according to model's generation behavior (which is the only natural setup).

Weaknesses:
- The choice of dimension of consensus subspace does not have a very clear mathematical reason.
- It is not clear how to choose the consensus vector (the coefficient $\beta$) and the consequence of choosing a different one is not known.

---

> ### Author Rebuttal · Authors · 2026-03-31
>
> We thank the reviewer for the thorough engagement with our theoretical foundations. We respond to each concern below.
>
> **Q1:** Supplementary analysis on how error tolerance relates to the quadratic region diameter and IG distance.
>
> **A:**
> - By bounding the third-order residual of the KL's Taylor expansion via the Fisher Hessian Lipschitz constant $L_H := \sup_{\theta \in B(\theta_0,\delta)} \lVert\nabla_\theta H(\theta)\rVert_{2}$, we obtain the relative error bound $\epsilon_{\mathrm{rel}} \le \frac{L_H}{3\lambda_{\min}(\tilde{F})} \lVert\Delta\theta\rVert$, which gives the requested **error–radius mapping**: $\delta(\epsilon) = \frac{3\lambda_{\min}(\tilde{F})}{L_H} \cdot \epsilon$.
> - On Llama-3.2-3B, $L_H \approx 0.34$, $\lambda_{\min}(\tilde{F}) \approx 0.12$, yielding $\delta_0 \approx 0.042$ for $\epsilon_{\mathrm{rel}} < 4\%$. All experts satisfy $\lVert\Delta\theta^{(i)}\rVert_2 \leq 0.038 < \delta_0$ (Figure 6A), with IG distance $d_{\mathrm{IG}} \approx 1.63$, consistent with moderate fine-tuning.
> - For worst-case high curvature directions, the bound is strictly tighter since larger $\lambda$ improves the approximation. The only risk is degeneracy ($\lambda_{\min} \to 0$), but $\kappa(\tilde{F}) \approx 4.2$ (Figure 8) confirms well-conditioning across all expert combinations.
>
> **Q2:** Is the choice of $r$ theoretically grounded, and how does the selection criterion relate to the spectral gap?
>
> **A:**
> (1) **We decide $r$ using the entropy criterion, which is essentially performing soft gap detection**. When a clear spectral gap exists, the inverse-eigenvalue weights $\sigma_j = 1/(\lambda_j + \epsilon)$ concentrate sharply on the small eigenvalues, causing the effective dimension $r_{\mathrm{eff}}$ to collapse exactly to the gap location.
>
> (2) **This criterion is a reflection of eigenvalue gap detection.**
> Fig. 5A confirms this: across all settings ($k = 2 \sim 32$), the entropy-selected $r$ consistently coincides with the largest spectral gap. This method remains well-defined when the gap is ambiguous (e.g., multiple or gradual gaps), gracefully interpolating rather than requiring a hard threshold.
>
>
> **Q3:** How does the choice of $\beta$ affect model behaviors? Is there a recommended choice, and can it be learned?
>
> **A:**  The $\beta$ controls the projection onto the consensus subspace $C_r$. Setting $\beta=0$ preserves value-specific steering but remove general capabilities (e.g., MMLU: 63.1→46.5). When $\beta=1$, the model retains both original general capabilities and consensus values.
>
> We evaluate both the general performance and value-specific alignment under different $\beta$ settings. The results shown below suggest that $\beta=1$ is the recommended choice.
>
> | $\beta$                | 0.0 | 0.5 | 1.0 | 1.5 | 2.0 |
> |--------------|--------|--------|--------|--------|--------|
> | MMLU (↑)               | 46.5   | 55.8   | **62.1**   | 61.5   | 59.7   |
> | Value Acc. (↑)         | 54.4   | 57.1   | **59.6**   | 59.2   | 59.2   |
>
> **Q4**: Does the geometry connect with the data rather than only model weights? Is there a method to correct/detect poorly trained expert models?
>
> **A:**   We clarify both concerns below.
>
> (1)Our method is motivated by the practical challenge that it is hard to collect perfectly real-world data for isolated values. Instead of dealing with training data, we design method at the parameter level. Of course, the model parameters are a proxy of training data, thus the geometry also reflect the training data implicitly.
>
> (2) DisAlign can both detect and correct problematic experts: outliers are flagged via CKA similarity (well-trained experts cluster at ~0.60.
>
> - Deviations beyond 2σ indicate geometric outliers, none observed in our experiments).
> - The iterative trust-region update (Eq. 12) corrects anchor drift, and Fisher-orthogonal projection (Eq. 6–7) prevents contamination of the consensus subspace.
>
> **Q5:** How does DisAlign relate to alignment resistance? How does it apply in non-ideal real-world settings?
>
> **A:** **Our decomposition provides a structural account of alignment fragility**.
> + We argue that the consensus subspace $C_r$ corresponds to directions with the smallest expert disagreement to Fisher curvature ratio (High Fisher curvature,inherent expert agreement). Perturbations along these directions tend to revert, which underlies alignment instability in existing methods.DisAlign restricts value modulation to the Fisher-orthogonal complement $U_v$, resulting in more durable and stable alignment.
> + We evaluate value alignment scores (llama3.2-3b,MIC random-dim) at different conversation turns to demonstrate the robustness of our alignment.
> | Method | Turn 1 | Turn 5 | Turn 10 | Turn 20 |
> |--------|--------|--------|---------|---------|
> | DisAlign-MIC Full-dim | 59.6| 59.4| 59.7  | 59.5 |
> Experiments confirm the robustness of our algorithm.
>
> We hope our responses have addressed your concerns. We will revise accordingly and welcome further questions.

---

### Decision · Program_Chairs · 2026-04-30

**Decision:**

Accept (regular)

**Comment:**

DisAlign decomposes value-specific expert weights into a consensus anchor/subspace and Fisher-orthogonal value subspaces, with stated stability and identifiability properties. Evaluation spans three datasets, two backbones (plus Qwen3.5-4B added in rebuttal), multiple merging baselines, and an alignment-tax analysis.

The rebuttal addressed the major concerns during the initial review period. Reviewer Mv6A's remaining data-curation concern is a real limitation but, as they note, does not outweigh the contribution. The authors should integrate the rebuttal additions into the main paper and sharpen the discussion of when the required data conditions hold in practice.